# A blood-based biomarker workflow for optimal tau-PET referral in memory clinic settings

Wagner S. Brum [1,2,20], Nicholas C. Cullen [3,4,20], Joseph Therriault [5,6,20], Shorena Janelidze[3], Nesrine Rahmouni[5,6], Jenna Stevenson[5,6], Stijn Servaes[5,6], Andrea L. Benedet[1], Eduardo R. Zimmer [2,5,7,8], Erik Stomrud[3,9], Sebastian Palmqvist [3,9], Henrik Zetterberg [1,10,11,12,13,14], Giovanni B. Frisoni[15], Nicholas J. Ashton[1,16,17,18], Kaj Blennow [1,10], Niklas Mattsson-Carlgren[3,4,19], Pedro Rosa-Neto [5,6] & Oskar Hansson [3] ✉

Blood-based biomarkers for screening may guide tau positrion emission tomography (PET) scan referrals to optimize prognostic evaluation in Alzheimer's disease. Plasma Aβ42/Aβ40, pTau181, pTau217, pTau231, NfL, and GFAP were measured along with tau-PET in memory clinic patients with subjective cognitive decline, mild cognitive impairment or dementia, in the Swedish BioFINDER-2 study (n = 548) and in the TRIAD study (n = 179). For each plasma biomarker, cutoffs were determined for 90%, 95%, or 97.5% sensitivity to detect tau-PET-positivity. We calculated the percentage of patients below the cutoffs (who would not undergo tau-PET; "saved scans") and the tau-PET-positivity rate among participants above the cutoffs (who would undergo tau-PET; "positive predictive value"). Generally, plasma pTau217 performed best. At the 95% sensitivity cutoff in both cohorts, pTau217 resulted in avoiding nearly half tau-PET scans, with a tau-PET-positivity rate among those who would be referred for a scan around 70%. And although tau-PET was strongly associated with subsequent cognitive decline, in BioFINDER-2 it predicted cognitive decline only among individuals above the referral cutoff on plasma pTau217, supporting that this workflow could reduce prognostically uninformative tau-PET scans. In conclusion, plasma pTau217 may guide selection of patients for tau-PET, when accurate prognostic information is of clinical value.

Tau pathology (neurofibrillary tangles and dystrophic neurites) is a key hallmark of Alzheimer's disease (AD) and has become an important focus in research, clinical trials, and clinical practice, partly due to improvements in methods to detect tau pathology in vivo through positron emission tomography (PET)[1,2]. Tau-PET is strongly associated with cognitive function, and can more accurately predict future cognitive decline than e.g., amyloid-β (Aβ) PET, MRI, and plasma AD biomarkers[3,4]. Tau-PET can also detect different clinically relevant subtypes of AD[5], and has been shown to improve clinical management of cognitively impaired patients even after thorough clinical and biomarker phenotyping[6]. Considering its clinical potential and its relatively recent approval for clinical use by the U.S. Food and Drug Administration, its use in clinical practice is expected to rise[7]. However, PET scans are relatively expensive, involve exposure to radiation, and timeslots on PET scanners are often restricted. Upon clinical implementation, unselected use of tau-PET in memory clinics is likely to

result in many avoidable scans in patients unlikely to harbor tau pathology. Thus, a funneled workflow to optimize the referral of patients who would benefit from tau-PET would be valuable, if it could minimize the number of scans while still preserving the diagnostic and prognostic properties of tau-PET in clinical practice[8].

Specific tau biomarkers in blood accurately detect tau pathology measured through tau-PET[9,10]. These include different forms of phosphorylated tau such as pTau181[11–13], pTau217[13–18], and pTau231[13,19], while other biomarkers such as GFAP[20,21], neurofilament light[22], and amyloid[23] may be useful for detecting AD-related changes up- or downstream of tau accumulation. These advances in blood-based tau biomarkers suggest that it may be possible to identify patients with cognitive complaints at higher risk of tau pathology, and thereby avoid tau-PET in patients who are at low risk for tau pathology and would not benefit from such a scan.

Previous studies evaluating screening for amyloid-PET risk using blood-based biomarkers have shown promise[24–26], but it is still unclear whether blood markers can be used to optimize the use of tau-PET in patients with cognitive symptoms where prognostic information is essential. Thus, we studied whether plasma biomarkers could be used for screening in a clinical setting to reduce the total number of tau-PET scans while still performing tau-PET in patients that might be tau-positive, and who could benefit from tau-PET to determine tangle deposition burden and spatial patterns. First, we evaluated a comprehensive panel of blood-based biomarkers to screen for tau-PET positivity in two memory clinic-based cohorts (BioFINDER-2 and TRIAD) in patients with subjective cognitive decline, mild cognitive impairment, or any form of dementia. We grounded our analysis based on the idea that patients likely to have abnormal tau accumulation should not be denied from tau-PET due to blood-based biomarker screening. We, therefore, focused on the ability of blood-based biomarkers to reduce negative tau-PET scans while also maintaining a high level of sensitivity so that >90% of tau-PET positive patients would receive a tau-PET scan according to the proposed workflow. Finally, in BioFINDER-2, we determined the prognostic value of tau-PET in those who were screened as negative or positive, to verify that tau-PET only provided prognostic information in those with abnormal plasma biomarker results (i.e., only in those who should undergo tau-PET according to the plasma biomarker prescreening).

## Results

### Participant characteristics

A total of 548 participants in the BioFINDER-2 cohort met the criteria to be included in the present analysis based on the availability of at least one of the plasma biomarkers listed below and a tau-PET scan. The average age of the cohort was 71.0 ± 8.6 years, with 45% of participants being female, and an overall tau-PET positivity rate of 37% (see Methods for details). Participants had subjective cognitive decline ($n = 135$), mild cognitive impairment ($n = 181$), or different forms of dementia ($n = 225$)—specifically AD dementia ($n = 140$), bvFTD ($n = 16$), svPPA ($n = 3$), nfvPPA ($n = 1$), DLB ($n = 24$), dementia with Parkinson's disease ($n = 7$), vascular dementia ($n = 16$) and unclassified dementia ($n = 24$). Among TRIAD participants meeting the inclusion criteria ($n = 179$), patients had a mean age of 70.0 ± 7.7 years and 57% were female, with a tau-PET positivity rate of 44%. Participants had SCD ($n = 23$), MCI ($n = 72$), AD dementia ($n = 58$), and non-AD dementias ($n = 26$) such as bvFTD ($n = 9$), svPPA ($n = 1$), vascular dementia ($n = 6$). Participant characteristics are presented fully in Table 1.

### Plasma biomarker screening and reduced PET scans

First, we tested how many tau-PET scans that could be saved in memory clinic settings by using either plasma Aβ42/Aβ40, pTau181, pTau217, pTau231, NfL, or GFAP to screen out patients at sufficiently low-risk, while still performing scans in those likely to have positive scans. As expected, there was a tradeoff between the percentage of total tau-PET scans saved and the sensitivity for tau-PET positivity (Fig. 1A). At 95% sensitivity (i.e., theoretically missing 5% of positive tau-PET scans), the percentage of tau-PET scans saved ranged from 11.1 to

**Table 1 | Cohort characteristics**

| | BioFINDER-2 | | | | | TRIAD | | | | |
|---|---|---|---|---|---|---|---|---|---|---|
| | **SCD** (N = 135) | **MCI** (N = 181) | **AD** (N = 140) | **Non-AD** (N = 85) | **Overall** (N = 548) | **SCD** (N = 23) | **MCI** (N = 72) | **AD** (N = 58) | **Non-AD** (N = 26) | **Overall** (N = 179) |
| Age, years | 66.6 (8.9) | 70.9 (8.7) | 73.8 (7.0) | 73.5 (7.2) | 71.0 (8.6) | 73.6 (5.8) | 72.3 (5.8) | 67.0 (8.3) | 67.5 (9.4) | 70.0 (7.7) |
| Female, n (%) | 65 (48.1) | 80 (44.2) | 72 (51.4) | 29 (31.5) | 244 (45.1) | 16 (69.6) | 38 (52.8) | 31 (53.4) | 17 (65.4) | 102 (57.0) |
| Education, years | 12.9 (3.9) | 12.6 (4.2) | 12.2 (4.3) | 11.4 (3.8) | 12.4 (4.1) | 16.7 (3.9) | 15.2 (3.8) | 14.5 (3.6) | 14.6 (4.0) | 15.1 (3.8) |
| MMSE score | 28.7 (1.4) | 27.0 (1.9) | 20.3 (4.3) | 22.6 (4.4) | 25.0 (4.6) | 25.2 (5.9) | 29.3 (0.89) | 28.0 (1.9) | 19.9 (6.3) | 25.3 (6.5) |
| *APOE* ε4 carriers, n (%) | 70 (51.9) | 100 (55.2) | 99 (70.7) | 36 (39.1) | 303 (56.0) | 10 (43.5) | 31 (43.1) | 32 (55.2) | 3 (11.5) | 76 (42.5) |
| Tau-PET positive, n (%) | 10 (7.4) | 57 (31.5) | 125 (89.3) | 9 (9.8) | 200 (37.0) | 1 (4.3) | 28 (38.9) | 49 (84.5) | 1 (3.8) | 79 (44.1) |
| Aβ-positive, n (%)* | 59 (43.7) | 102 (56.4) | 136 (97.1) | 38 (41.3) | 333 (61.6) | 6 (26.1) | 49 (68.1) | 51 (87.9) | 3 (11.5) | 109 (60.9) |
| Plasma Aβ42/Aβ40** | 0.81 (0.04) | 0.81 (0.03) | 0.82 (0.03) | 0.80 (0.03) | 0.81 (0.03) | 0.14 (0.17) | 0.09 (0.03) | 0.11 (0.06) | 0.12 (0.08) | 0.11 (0.09) |
| Plasma pTau181, pg/mL** | 5.5 (7.6) | 5.7 (7.1) | 7.0 (4.5) | 5.2 (6.1) | 5.9 (6.6) | 11.3 (6.2) | 14.6 (7.3) | 23.5 (10.4) | 13.9 (14.2) | 16.8 (10.6) |
| Plasma pTau217, pg/mL** | 1.4 (1.6) | 2.9 (4.3) | 7.6 (3.9) | 2.1 (2.4) | 3.7 (4.2) | 0.07 (0.04) | 0.12 (0.07) | 0.26 (0.16) | 0.074 (0.09) | 0.15 (0.13) |
| Plasma pTau231, pg/mL | 11.2 (4.7) | 13.1 (7.1) | 18.0 (7.5) | 12.8 (5.8) | 13.8 (6.9) | 14.0 (6.6) | 17.8 (9.1) | 24.8 (11.9) | 15.8 (10.1) | 19.3 (10.7) |
| Plasma NfL, pg/mL | 13.9 (6.9) | 20.0 (17.2) | 27.1 (25.6) | 26.8 (13.2) | 21.5 (18.3) | 34.0 (33.9) | 24.6 (13.0) | 31.2 (10.8) | 30.7 (18.7) | 28.9 (17.6) |
| Plasma GFAP, pg/mL | 205 (166) | 224 (109) | 393 (180) | 259 (148) | 268 (168) | 260 (114) | 278 (150) | 370 (149) | 204 (168) | 294 (158) |

This table displays the characteristics of participants in the present analysis. Continuous variables are reported as mean and standard deviation, while categorical variables are reported as total counts and percentages per category in the study population. All plasma biomarker levels are reported in pg/ml. In BioFINDER-2, the non-AD diseases represented in the cohort include the following: bvFTD (n = 16), svPPA (n = 3), nfvPPA (n = 1), Lewy body dementia (n = 24), PDD (n = 7), vascular dementia (n = 16) and unclassified dementia (n = 24). In TRIAD, the non-AD diseases represented in the cohort include the following: bvFTD (n = 9), svPPA (n = 1), PDD (n = 1), vascular dementia (n = 6), corticobasal syndrome (n = 1), progressive supranuclear palsy (n = 3), cerebral amyloid angiopathy (n = 1) and unclassified dementia (n = 4).
*Aβ-status was determined via CSF Aβ42/Aβ40 in BioFINDER-2 and with 18F-AZD4694 in TRIAD.
**Different assay versions for this biomarker are used in each cohort as described in the methods.

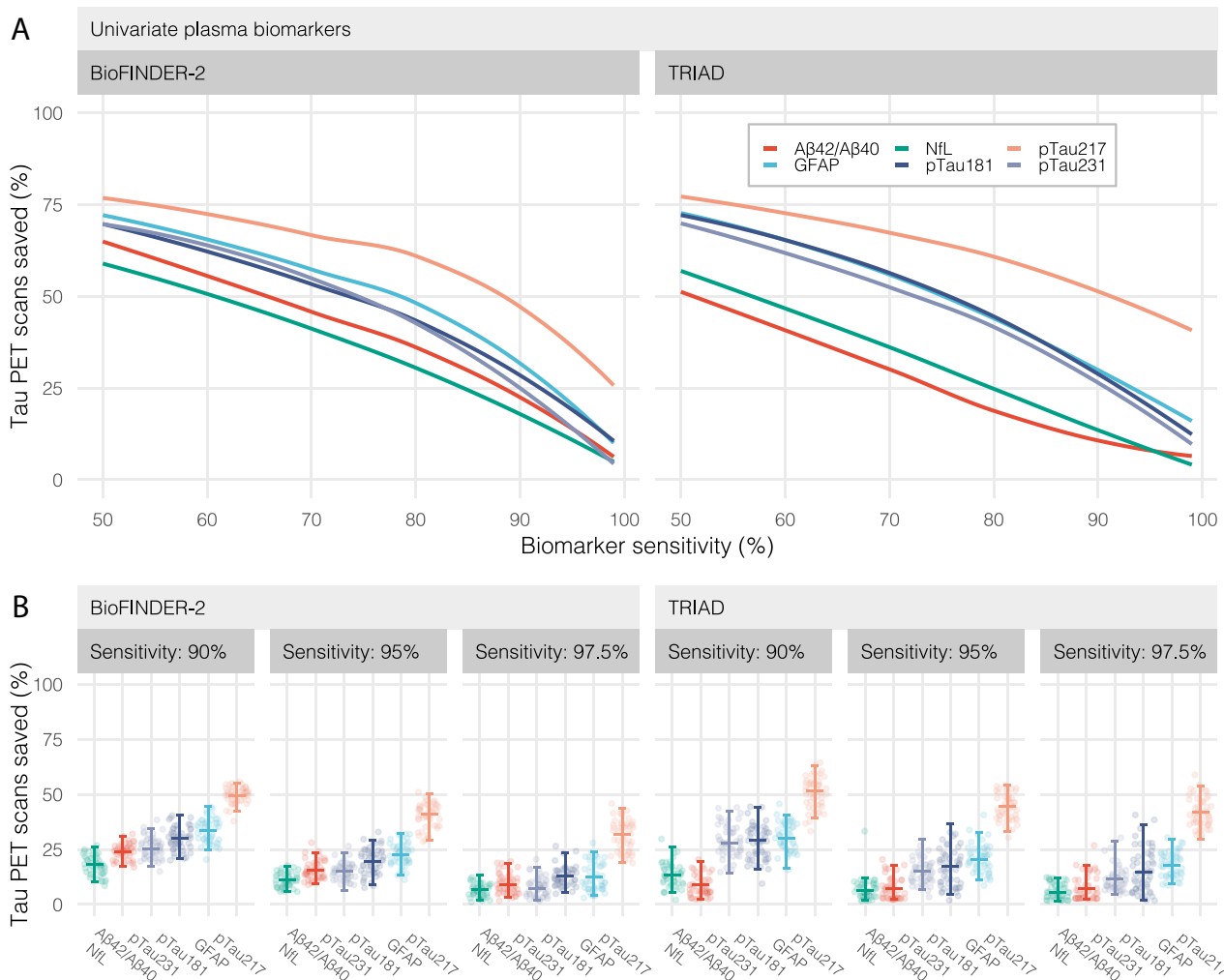

**Fig. 1 | Tau-PET scans saved as a function of biomarker sensitivity. A** The top panel shows the percentage of tau-PET scans saved over the range of possible biomarker sensitivities for detecting tau-PET positive individuals for the BioFINDER-2 (left; *n* = 548) and TRIAD (right; *n* = 179) cohorts. Lines were estimated using a "loess" function over all bootstrapped trials. For example, a biomarker cutoff for pTau217 set at 90% sensitivity would result in a 50% reduction in tau-PET

scans. **B** The lower panels show the percentage of tau-PET scans at specific bio-marker sensitivity values (90, 95, and 97.5%) with confidence intervals and median derived from 100 bootstrap trials and represented by solid color bars. Source data are provided as a Source Data File. PET positron emission tomography, Aβ amyloid-beta, pTau phosphorylated tau, NfL neurofilament light, GFAP glial fibrillary acidic protein.

42.3% for different plasma biomarkers in BioFINDER-2 and from 5.4 to 44.6% in TRIAD. Notably, plasma pTau217 screening led to significantly higher percentages of reduced tau-PET scans (BioFINDER-2: 42.3%; TRIAD: 44.6%) compared with all other biomarkers (*P* < 0.001 for all comparisons in both cohorts), followed by GFAP (BioFINDER-2: 22.5%; TRIAD: 20.2%), pTau181 (BioFINDER-2: 19.4%; TRIAD: 16.7%), pTau231 (BioFINDER-2: 15.3%; TRIAD: 16.7%), Aβ42/Aβ40 (BioFINDER-2: 14.8%; TRIAD: 5.9%), and NfL (BioFINDER-2: 11.1%; TRIAD: 5.4%) (Fig. 1B). A similar ranking of blood-based biomarkers was observed when evalu-ating a more lenient screening strategy with sensitivity at 90% or a more stringent strategy with 97.5% sensitivity. At 90% sensitivity (i.e., missing 10% of tau-PET-positive patients), screening with pTau217 saved 50.7 and 52.1% of tau-PET scans in BioFINDER-2 and TRIAD, respectively. At 97.5% sensitivity, the percentage reduction in scans achieved by pTau217 screening was 33.1 and 41.6% in BioFINDER-2 and TRIAD, respectively. At 90% sensitivity, other biomarkers saved 15.8–31.9% (BioFINDER-2) and 8.1–52.1% (TRIAD) of tau-PET scans with GFAP and pTau181 following pTau217 (31.9 and 29.5% saved scans). At 97.5% sensitivity, other biomarkers' ability to save tau-PET scan ranged from 6.5 to 11.8% (BioFINDER-2) and from 5.9 to 13.6% (TRIAD).

In a sensitivity analysis, we evaluated whether adding age, sex, and *APOE* ε4 status to each plasma biomarker would increase the percen-tage of tau-PET scans saved. In both TRIAD and BioFINDER-2, this generally led to slight increases in saved scans for all biomarkers, but none of these increases were significant (Supp. Fig. 1).

In BioFINDER-2, we also conducted a sensitivity analysis in which the same univariate screening strategies were evaluated but with the outcome as a positive tau-PET scan determined by a novel clinically validated method for visual interpretation of the RO1498 tau tracer[6]. The results were generally similar, with pTau217 performing best, fol-lowed by GFAP and pTau181 in a second tier, with pTau231, Aβ42/Aβ40, and NfL performing worse in terms of saved scans (Supp. Fig. 2). For example, screening for a visually positive tau-PET scan with plasma pTau217 reducing 50.2% scans at 90% sensitivity, 41.6% at 95% sensi-tivity and saving 30.0% of scans at 97.5% sensitivity.

**Rate of tau-PET positivity at different screening sensitivities**
Next, positive predictive value (PPV) was calculated as the percentage of tau-PET-positive individuals among those who would be selected for tau-PET according to their plasma biomarker results (Fig. 2). In both

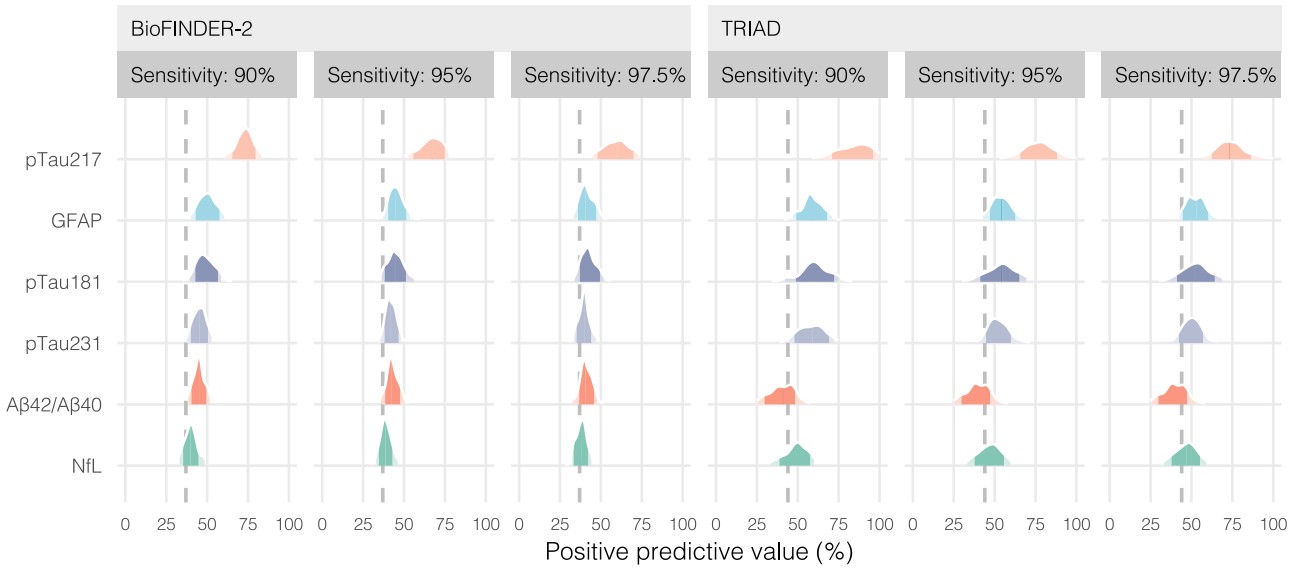

**Fig. 2 | Positive predictive value at different sensitivity thresholds.** This figure shows the positive predictive value (PPV) of the individual plasma biomarkers at different sensitivity thresholds (90, 95, and 97.5%) for the BioFINDER-2 (left; *n* = 548) and TRIAD (right; *n* = 179) cohorts. Here, PPV represents the percentage of true positive tau-PET scans that can be expected among those individuals who are selected for PET scanning via plasma biomarker screening. Conversely, 1 - PPV represents the percentage of individuals who receive a tau-PET scan (due to high risk for tau pathology as predicted by each plasma biomarker) but are expected to actually be negative for tau pathology. The dotted lines represent the tau-PET positivity rate for BioFINDER-2 (37%) and TRIAD (44%), with pTau217 being the only biomarker providing clear increases in relation to the population positivity rates in both cohorts. Source data are provided as a Source Data File. PET positron emission tomography, Aβ amyloid-beta, pTau= phosphorylated tau, NfL neurofilament light, GFAP glial fibrillary acidic protein.

BioFINDER-2 and in TRIAD, the only plasma biomarker to show a PPV significantly ($p < 0.05$) greater than the study population rate of tau-PET positivity (BioFINDER-2: 37.0%; TRIAD: 44.1%) at all evaluated strategies (90, 95 and 97.5% sensitivity) was pTau217. At 95% sensitivity, the PPVs ranged from a 1.8–30.3 percentage point increase over the population rate in BioFINDER-2 and from 3.1 to 33.1% in TRIAD (except for Aβ42/Aβ40, which led to −4.0%). These increases over the population rate at this sensitivity (95%) were only significant for pTau217 and GFAP in BioFINDER-2 and only for pTau217 in TRIAD. At 95% sensitivity, pTau217 significantly outperformed all other biomarkers ($p < 0.0001$), giving a PPV of 67.3% (95% CI 52.6–77.0%) in BioFINDER-2 and of 77.1% (95% CI 60.1–94.6%) in TRIAD. In both cohorts, similar results favoring pTau217 were seen for screening cutoffs at 90%, with PPVs of 73.1% in BioFINDER-2 and of 86.2% in TRIAD, or when using screening cutoffs at 97.5% sensitivity, with PPVs of 60.1% in BioFINDER-2 and 73.1% in TRIAD.

When repeating the PPV analysis using tau-PET positivity determined with the visual read method in BioFINDER-2, pTau217 was again the only biomarker to lead to significant increases over the population rate of visual tau-PET positivity (45.3%) (Supp. Fig. 3). The tau-PET scans were considered positive when abnormal accumulation occurred either confined to the temporal lobe (early deposition pattern) or when accumulation occurred in areas outside the temporal lobe (e.g., frontal, parietal, occipital cortices; late deposition pattern). At 95% sensitivity, pTau217, showed a very similar PPV (66.5%, 95% CI 53.9–77.0%; 21.2% increase over population rate) as that of the quantitative-cutoff analysis above. At 90% and 97.5% sensitivities, pTau217 showed PPV's of 74.0% (95% CI 63.5–81.9%) and 57.3% (95% CI 46.2–70.9%), respectively.

Full details regarding PPVs, NPVs, and saved scans are presented for all biomarkers in Supp. Table 1 (BioFINDER-2; quantitative cut-off), 2 (TRIAD), and 3 (BioFINDER-2; visual read).

### Subgroup analysis in SCD-MCI and all-cause dementia
Next, we determined whether the usefulness of plasma biomarker screening differed between non-demented and demented patients in BioFINDER-2 (Supp. Fig. 4) and TRIAD (Supp. Fig. 5). In BioFINDER-2 and TRIAD, there was no significant change in the percentage of tau-PET scans that could be saved via blood-based biomarker screening when looking at non-demented patients alone (SCD-MCI) and requiring 90 or 95% sensitivity to detect tau-PET positive individuals. In contrast, both pTau217 and GFAP saw a significant decrease in the percentage of saved tau-PET scans when requiring 90% or 95% sensitivity and looking at the all-cause dementia group alone in BioFINDER-2, with a similar trend observed for pTau217 in TRIAD.

### Tau-PET load in screened-in and screened-out patients
To characterize the screened-in and screened-out populations based on pTau217, the best-performing biomarker in our main analysis, we show in Fig. 3 the tau-PET SUVr values for those with normal vs elevated plasma pTau217 at the 90, 95, and 97.5% sensitivity cutoffs derived in the previous sections in BioFINDER-2 (Fig. 3; top) and TRIAD (Fig. 3; bottom). In general, very few patients had elevated tau-PET retention in the group with normal pTau217 values in BioFINDER-2 (*n* = 19 [7.8%], *n* = 11 [5.1%], *n* = 6 [2.9%] for 90, 95, and 97.5% sensitivity-based cutoffs, respectively) and TRIAD (*n* = 7 [7.9%], *n* = 3 [4.2%], *n* = 1 [1.5%] for 90, 95, and 97.5% sensitivity-based cutoffs, respectively). The figure also shows that less stringent strategies, such as a sensitivity of 90%, would lead to greater reductions in tau-PET scans, while more stringent strategies, such as 95 and 97.5% sensitivities, may have better potential to minimize false-positives blood biomarker results. In the Supplementary Information, we show that the 95% sensitivity plasma p-tau217 screening strategy also led to reliable classification of BioFINDER-2 tau-positive patients based on visual read, especially those with an advanced deposition pattern (Supp. Fig. 6).

### Tau-PET prognostic value in screened-in and screened-out patients
Finally, we determined whether our suggested approach consisting of screening with plasma pTau217, followed by tau-PET only in those with elevated plasma pTau217, would be valuable when predicting cognitive decline in the BioFINDER-2 cohort (prognostic analyses not

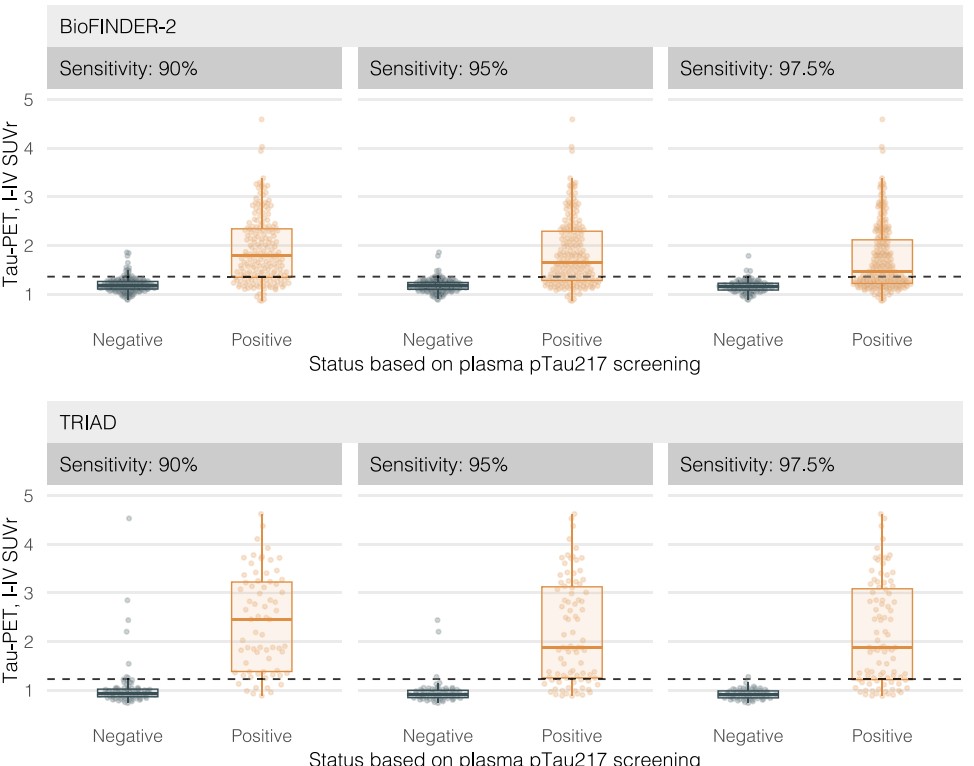

**Fig. 3 | Tau-PET values in screened-in versus screened-out patients based plasma pTau217.** This figure shows the tau-PET levels according to plasma pTau217 levels based on different screening strategies (Sensitivities; left: 90%, middle: 95%, right: 97.5%). The y-axis dots indicate the standardized uptake value ratio (SUVr) based on a Braak I-IV meta-temporal composite region. The x-axis indicates whether patients had abnormal values according to the cutoff for each strategy.

Boxplots represent the median, interquartile range (IQR), and range to smallest or largest value no further away from 1.5*IQR. Results for BioFINDER-2 ($n = 402$) are presented in the top row, and in the bottom row for TRIAD ($n = 160$). Source data are provided as a Source Data File. PET positron emission tomography, pTau phosphorylated tau, SUVr standardized uptake value ratio.

performed in TRIAD; see Methods). We determined (i) whether tau-PET is superior to demographic variables and plasma pTau217 to predict cognitive decline in those with elevated plasma pTau217 levels (to determine whether tau-PET has any added value in this pTau217-positive population), and (ii) whether tau-PET has no or minor predictive value in those patients with low plasma pTau217 levels (who would not be selected for tau-PET). We evaluated the prediction of annual change in MMSE separately in patients who had pTau217 levels above or below the 95% sensitivity cutoff calculated in the primary analysis and compared four models for each population: (i) demographics-only (age and sex); (ii) plasma pTau217 and demographics; (iii) tau-PET and demographics; (iv) tau-PET, plasma pTau217, and demographics. We compared models using $R^2$, which indicates how well the model explains outcome variability (the higher, the better), and using weighted Akaike Information Criteria (wAIC), a metric ranging from 0 to 1 indicating which model, among a set of compared nested models, provides the best tradeoff between goodness-of-fit and model complexity (wAIC of compared models sums to 1, and the higher wAIC, the higher probability that the model provides the best trade-off).

In patients with SCD or MCI with abnormal pTau217 levels (who would be referred to a tau-PET scan in our proposed workflow) (Fig. 4A), the model containing tau-PET SUVr values and demographics had the highest added value for predicting cognitive decline ($R^2 = 29.8\%$; wAIC = 0.67) and performed significantly better when compared to the demographics-only model ($R^2 = 5.1\%$; wAIC = 0) and the plasma pTau217 plus demographics model ($R^2 = 13.8\%$; wAIC = 0). While performing similarly to the full model containing plasma pTau217, tau-PET, and demographics ($R^2 = 29.7\%$; wAIC = 0.33), the model containing tau-PET and demographics was favored due to being

more parsimonious. In contrast, a tau-PET scan would not add prognostic information when evaluating annual MMSE change in the patients with SCD or MCI below the pTau217 cutoff (these patients would not be selected for tau-PET in our suggested workflow). In this group, the demographic model performed best based on being most parsimonious ($R^2 = 11.3\%$; wAIC = 0.43) when compared to the plasma pTau217 plus demographics-only ($R^2 = 11.5\%$; wAIC = 0.28), tau-PET plus demographics ($R^2 = 10.9\%$; wAIC = 0.18) and the full model containing tau-PET, pTau217 and demographics ($R^2 = 11.0\%$; wAIC = 0.11) models.

Similarly, when predicting annual MMSE change in all-cause dementia patients (Fig. 4B) above the pTau217 cutoff, the model containing tau-PET SUVr and demographics showed the highest added prognostic value and performed significantly better ($R^2 = 25.6\%$; wAIC = 0.75), when compared to the demographics-only ($R^2 = 0.2\%$; wAIC = 0) and plasma pTau217 plus demographics ($R^2 = 7.7\%$; wAIC = 0) models. Again, the tau-PET plus demographics performed similarly to the full model containing tau-PET, plasma pTau217, and demographics ($R^2 = 25.0\%$; wAIC = 0.25) model, but was favored for being more parsimonious. In participants below the pTau217 cutoff, none of the models was associated with annual change in MMSE ($R^2 = 0$ for all). Raw trajectories for these subpopulations are shown in Supplementary Information (Supp. Fig. 7).

## Discussion
In two independent cohorts, we found that AD plasma biomarkers, especially plasma pTau217, can be used to pre-screen patients with cognitive complaints to identify those where tau-PET is unlikely to provide important information. Specifically, we found that a cutoff of plasma pTau217 with 95% sensitivity for tau-PET positivity could

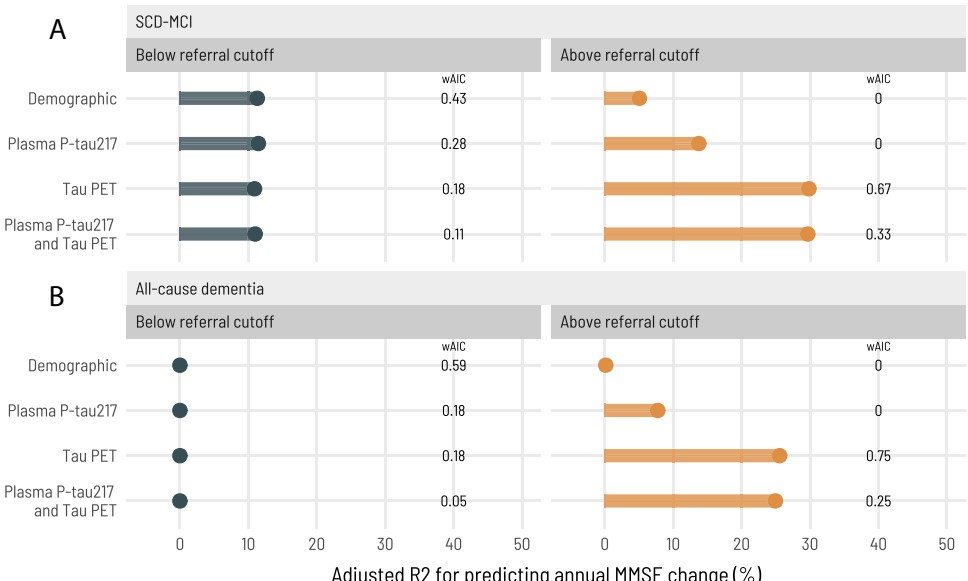

**Fig. 4 | Prognostic performance of tau-PET in individuals screened-in versus screened-out based on plasma pTau217 levels.** This figure shows the added prognostic value of tau-PET in predicting annual change in MMSE separately for individuals below (left panels; dark green) and above the referral cutoff (right panels; orange), based on the 95% sensitivity plasma pTau217 cutoff herein calculated, compared to a demographic model and a plasma pTau217 model. Each of the four models are represented in the y-axis, and the x-axis represents the adjusted $R^2$ of the models, indicating how well they predict annual change in MMSE. The weighted Akaike Information Criterion (wAIC), a measure from 0 to 1 with a higher value indicating which model is more likely to be correct among those tested, is indicated in vertically disposed black text. In both SCD-MCI (**A**) and all-cause dementia (**B**) clinical populations, a tau-PET scan was not useful for screened-out individuals (SCD-MCI: $n = 147$; All-cause dementia: $n = 30$), with the demographic model being more parsimonious; for those above the referral cutoff (SCD-MCI: $n = 101$; All-cause dementia: $n = 123$), the tau-PET-only model was significantly better than the other three models. Source data are provided as a Source Data File. PET positron emission tomography, pTau phosphorylated tau, R2 coefficient of determination, wAIC weighted Akaike Information Criterion, MMSE Mini-Mental State Exam, MCI mild cognitive impairment, SCD subjective cognitive decline.

reduce the number of tau-PET scans by almost half, while reaching a tau-PET-positivity rate of two-thirds among those who would be referred to a scan. Optimizing a referral workflow is important since tau-PET is an outstanding method for the prediction of future cognitive decline, detection of relevant AD subtypes, and clinical management improvement[4,5]. Importantly, in BioFINDER-2, we found that tau-PET only provided significant prognostic information in patients with high plasma pTau217 levels, but not in those with low pTau217 levels. Taken together, our results support a two-step workflow to optimize the prognostic clinical work-up of patients with cognitive complaints, with plasma pTau217 as a screening test (step 1) to identify those recommended to undergo tau-PET (step 2) (Fig. 5). Results were notably similar across the cohorts, and were very similar when using a visual read for tau-PET positivity determination in BioFINDER-2. This is reassuring, given that the visual read method has been validated to detect tau accumulation outside the temporal lobe, which further supports that our proposed workflow may not lead to a denial of scans for patients with atypical patterns of tau deposition.

We also quantified the percentage of negative tau-PET scans which could be saved at varying biomarker sensitivity thresholds and found that approximately half of tau-PET scans could be saved while achieving 90% or 95% detection sensitivity when using plasma pTau217. Even when requiring a very high pTau217 sensitivity (97.5%), more than 25% of uninformative tau-PET scans could be saved.

Another secondary goal was to quantify the PPV of tau-PET positivity among those who would be referred for a tau-PET scan based on blood-based biomarkers in screening. This metric is equal to the percentage of tau-PET positive scans that can be expected among memory clinic patients following the proposed blood-based biomarker screening workflow. In general, the PPVs were significantly greater than the study population prevalence of tau-PET positivity (~40%), which would correspond to the rate of tau-PET-positivity in the

unrealistic scenario of referring all patients to a tau-PET scan. PPVs would nearly double in comparison to the population prevalence in the best cases, suggesting a large improvement in the cost efficiency of tau-PET scanning. And when increasing the detection sensitivity threshold to 97.5%, the PPV naturally decreased because more tau-PET negative individuals would be screened-in based on the blood-based biomarker. When evaluating a more lenient strategy (90% sensitivity), the PPV among those screened-in would be the highest for pTau217 (73.1%), but at the expense of missing more tau-PET-positive patients at screening. Note that the observed PPVs for these cutoffs were derived to serve in a screening context, and are likely too low to support any cutoff-biomarker combinations as standalone tools for tau positivity in AD.

A strength of this study is that several state-of-the-art art biomarkers were included and compared. pTau217 performed better than all other biomarkers in terms of screening out tau-PET negative individuals while maintaining high sensitivity for detecting tau-PET positive individuals. By observing similar results in two cohorts with different pTau217 assays used, we show that the finding is generalizable to a different patient population and to different pTau217 assays. pTau181 and pTau231 were also quite effective at screening for tau-PET positivity. Results from many studies have agreed on the superior performance of plasma pTau217 compared to other pTau isoforms across multiple disease stages and outcomes[15,18,27,28]. Further, plasma pTau variants, especially pTau217, demonstrate higher disease-related fold changes, making them robust AD biomarkers and less susceptible to analytical variation[29–31]. Plasma pTau231 may react early to cerebral amyloid accumulation but plateau at a stage when pTau217 is dynamically changing and likely reflecting tau accumulation[32]. GFAP was also efficient to screen for tau-PET positivity, even at detection sensitivity levels of 95 and 97.5%, indicating a relationship between elevated GFAP and positive tau-PET status. Since GFAP has been

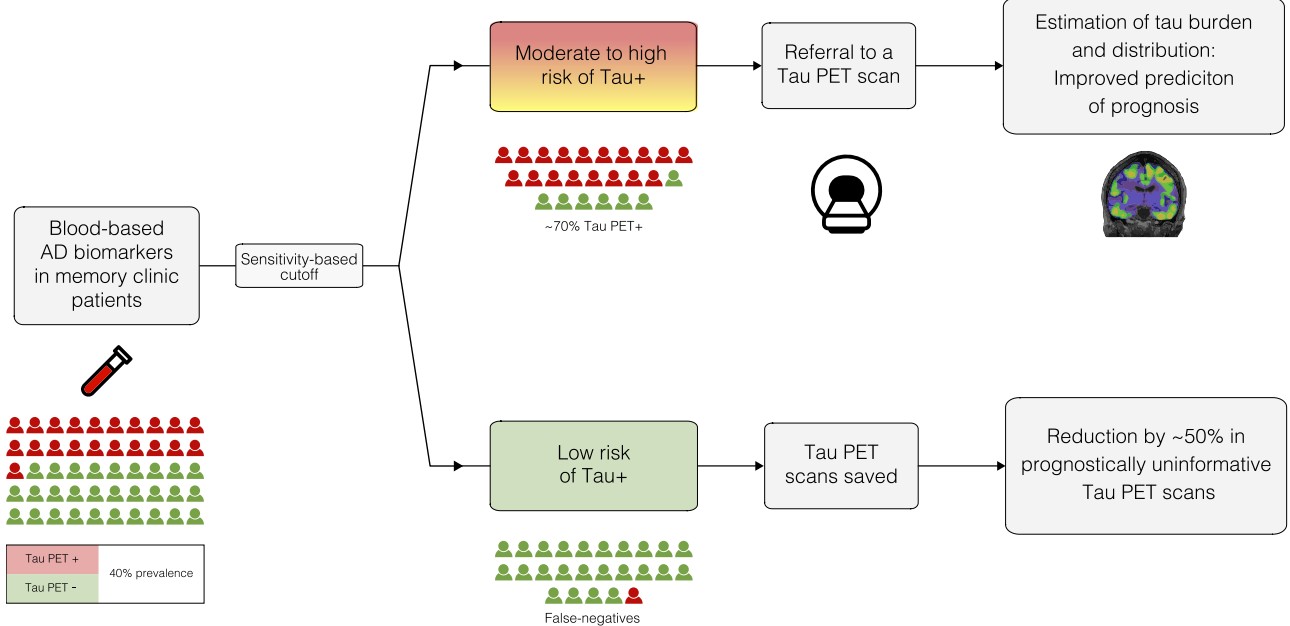

**Fig. 5 | Proposed screening workflow for optimizing tau-PET referrals in a memory clinic setting.** This figure shows a proposed workflow for the use of a blood-based biomarker algorithm for tau-PET screening in a memory clinic setting for optimizing prognostic assessment. The results from pTau217 – the highest-performing individual plasma biomarker—are demonstrated. In this example, ~50% of tau-PET scans would be avoided, while 95% of true tau-PET positive individuals would receive a tau-PET scan (i.e., 95% sensitivity), and there would be a ~70% tau-PET positive rate among those who receive a tau-PET scan (i.e., 70% positive predictive value). PET positron emission tomography, AD Alzheimer's disease.

growingly associated with Aβ pathology[20,21], it is possible that the performance observed here was partly due to GFAP's ability to screen out Aβ-negative individuals, who are mostly tau-negative.

We also found that Aβ42/Aβ40 and NfL led to increases in screening cost-efficiency as compared with not using any blood biomarker at all, but their increases in PPV compared with the population prevalence were the smallest. Even at a relatively low sensitivity threshold of 90%, these biomarkers would only save about 20% of tau-PET scans, while increasing the PPV by around only 10 percentage points as compared with the study prevalence. While they performed in a lower tier within our proposed application, these results do not affect the utility for diagnosis and prognosis of AD and neurodegeneration in general shown for Aβ42/Aβ40 and NfL in other contexts. An important next step in AD biomarker research is to investigate exactly how pTau and these other biomarkers can be combined to maximize their effects in specific applications[33].

Given the present results, it is important to highlight different aspects of the clinical value of tau-PET. First, a pre-screen with plasma pTau217 could rule out patients who are very likely tau-PET negative. Tau-PET could be avoided in those patients. However, we do not suggest to simply maximize the PPV of the screening method, since this would result in avoiding tau-PET also in many patients who would actually be tau-PET positive. When referring individuals with an intermediate or high probability of tau-PET positivity to a scan, a negative result would be crucial for differential diagnosis, while a positive scan would be key to determining the prognosis.

In our prognostic analyses, tau-PET only added value in predicting cognitive decline among patients above the screening cutoff of pTau217, in contrast to weak or no added value for those screened-out by pTau217. This highlights that the proposed two-step workflow would aid in referring the population that would benefit the most from having the severity of their tangle burden estimated. Tau-PET will still be very meaningful in these individuals, and will not just confirm tau positivity, but also give relevant information on topographical tau distribution, which is clinically useful per se and can also be used to

identify different AD subtypes with different clinical phenotypes and prognosis[5]. An analogy can be made to the context of prostate cancer screening, in which patients referred to advanced testing, e.g., biopsy, based on elevated plasma prostate-specific antigen levels need not only to have the presence of a neoplastic formation confirmed but also characterized based on prognostically relevant histopathological features[34].

Certain limitations with the present study exist. For one, looking at tau-PET status as a binary outcome (i.e., "negative" versus "positive") is not optimal from a statistical perspective. It might be more effective to establish a relationship between continuous blood-based biomarker status and continuous tau-PET levels. However, this approach still does not get around the fact that a threshold for tau-PET positivity must be established at some point for purposes of clinical decision-making. It is also worth mentioning that blood-based biomarker development is happening at a rapid pace—both in terms of identifying new target biomarkers, creating new assays, and refining existing assays—so claims about the superiority of one biomarker over another should be approached with caution. The plasma Aβ assay available for BioFINDER-2 was the Araclon assay and Simoa for TRIAD, which have not shown the best performance in recent head-to-head comparisons[35]. However, Aβ in the plasma presents a well-described robustness issue[30,31] and is not substantially associated with tau pathology[36], and has been recently shown to be affected by a common heart failure drug[37]. Thus, results would not likely change substantially had we used a different Aβ assay. Another limitation is that the tau-PET positivity prevalence depends on the case-mix in the specific memory clinic. While our screening process showed good results even in the SCD-MCI subgroup analyses (a population with a lower prevalence of tau-PET positivity and thus with lower PPVs for the studied diagnostic tests), the results were substantially better for the all-cause dementia population, which is compatible with the well-described fact that current tau blood biomarkers perform better in advanced stages. The BioFINDER-2 cohort consecutively includes patients in a secondary memory clinical with very few exclusion criteria to ensure a

representative population, with patients presenting relatively low educational attainment (mean 12 years), and a range of common age-related comorbidities: many included patients from BioFINDER-2 present age-related comorbidities as observed in the real-world populations of older adults, with 50.4% (n = 277) presenting with cardiovascular disease, 14.4% (n = 79) with diabetes and 34.4% (n = 189) with dyslipidemia. While race/ethnicity was not available for Bio-FINDER-2, among the 179 included TRIAD participants, 3 (1.7%) were Asian, 1 was black (0.5%), n = 172 were white (96%), and unknown/not reported for 3 (1.7%). Thus, further studies in more diverse populations are warranted. The overall prevalence of ~40% tau-PET positivity will affect the percentage of patients who are screened-out from taking a tau-PET scan, as well as the PPV among patients who end up undergoing a tau-PET scan. Lastly, prognostic analyses were carried out in BioFINDER-2 only, due to insufficient follow-up for the included participants from TRIAD. Simulating the effect of different theoretical tau-PET positivity prevalence values was not in the scope of this analysis, as changing prevalence assumptions without changing underlying biomarker levels can be misleading. Additional studies can contribute to this gap in knowledge.

To summarize, the results from this study suggest that blood-based biomarker screening can reduce uninformative tau-PET scans in memory clinic settings. Blood-based biomarker screening—especially using plasma pTau217—of patients with memory complaints may greatly reduce the overall number of tau-PET scans which are performed in the first place by screening out individuals at very low risk for being tau-PET positive. Blood-based biomarker screening with high-performing blood biomarkers like pTau217 can optimize the referral process of this costly but valuable PET method, ensuring that tau-PET scans are only performed in patients who will have their prognostic assessment benefited by the estimation of tangle burden and spatial deposition pattern.

## Methods
### Study design and participants
This manuscript used data from the BioFINDER-2 and TRIAD cohorts. BioFINDER-2 was approved by the Ethical Review Board in Lund, Sweden, which is part of the Swedish Ethical Review Authority (#2016-1053). TRIAD was approved by the Montreal Neurological Institute PET working committee and the Douglas Mental Health University Institute Research Ethics Board (IUSMD16-61, IUSMD16-60). All patients gave their written informed consent to participate in the study, and all participants were volunteers.

Participants from the Swedish BioFINDER-2 study (NCT03174938) who were classified as either subjective cognitive decline, mild cognitive impairment, or any form of dementia were included in the present analysis (see results for a breakdown). Among participants referred to the memory clinic not meeting the criteria for dementia as per DSM-5, those performing worse than −1.5 SD in any cognitive domain, based on age and education norms, were classified as having MCI. Those not meeting MCI criteria were classified as having SCD. For AD dementia, participants needed to meet DSM-5 criteria for dementia due to AD (major neurocognitive disorder) and present biomarker-evidence of Aβ-positivity as per the NIA-AA guidelines, with exceptions to this (e.g., borderline cut-off patients) being evaluated in an individual-case basis. Patients without typical AD clinical syndromes had a clinical diagnosis of fronto-temporal dementia, vascular dementia, Parkinson's disease, progressive supranuclear palsy, multiple systems atrophy, corticobasal syndrome, or primary progressive aphasia made based on appropriate clinical guidelines for each disorder[38–43].

We also included participants from the Translational Biomarkers in Aging and Dementia cohort (TRIAD), who were selected for having data on plasma biomarkers and tau-PET. These participants were sourced from a specialized tertiary care memory clinic dedicated to diagnosing and treating neurodegenerative diseases. This subset underwent comprehensive clinical evaluations, including the Clinical Dementia Rating (CDR), Mini-Mental State Examination (MMSE), and assessment of cerebrovascular disease risk using the Hachinski Ischemic Scale. Criteria for exclusion included uncontrolled systemic conditions not managed by stable medication, active substance misuse, recent significant head injuries, major recent surgeries, or contraindications for MRI/PET imaging.

Availability of plasma biomarker levels and a tau-PET scan was required for inclusion in the cross-sectional (BioFINDER-2 and TRIAD) and longitudinal analyses (BioFINDER-2 only), available data was also required for at least one follow-up visit with available data for the Mini-Mental State Exam (MMSE).

### Biomarker measurement
Plasma levels of pTau181, pTau217, pTau231, Aβ42/Aβ42, NfL, and GFAP were measured using different analytical platforms. Plasma Aβ42/Aβ42 was measured using a mass-spectrometry (MS) method from Araclon (BioFINDER-2) and Simoa (TRIAD)[35]. In BioFINDER-2, plasma pTau181 and pTau217 were measured using the MSD technology with immunoassays based on antibodies from Eli Lilly[11,15]. In TRIAD, pTau181 was measured with an in-house Simoa method (University of Gothenburg) and pTau217 with the Janssen assay[44]. Plasma pTau231 was measured using an in-house Simoa method at the University of Gothenburg[19]. Plasma NfL and GFAP were measured using commercial Simoa methods. Tau-PET was measured using [18F]-RO948 in BioFINDER-2 and [18F]-MK6240 in TRIAD, as previously described[45,46]. Abnormality was defined by an a priori-defined cutoff of 1.36 in BioFINDER-2 and of 1.24 in TRIAD based on a temporal meta-ROI[45,46]. In a sensitivity analysis, tau-PET positivity was determined based on a clinically validated visual read method[6]. Briefly, with this visual read method, patients are classified in (A) normal image; no discernible [18F]RO948 retention, (B) retention of [18F]RO948 confined to the temporal lobes, (C) more widespread retention of [18F]RO948, reaching into the parietal, occipital, or frontal lobes, and (D) inconclusive scan, and patients with the B and C patterns are herein considered as abnormal tau deposition. There was no considerable time lag between tau-PET scanning and blood collection in BioFINDER-2, and in TRIAD this lag was low (0.43 years on average).

### Statistical analysis
The primary analysis was to identify the highest cutoff value for each individual plasma biomarker which would achieve a sensitivity of 95% for identifying tau-PET positivity in the entire cohort (SCD + MCI + AD). A sensitivity of 95% implies that of the individuals who are truly tau-PET positive, only 5% would be classified by the plasma biomarker as likely to be tau-PET negative (and therefore miss a tau-PET scan). Other sensitivities (95 and 97.5%) were also tested.

Once these cutoffs were determined for each plasma biomarker, the percentage of individuals below the cutoff was calculated; this value represents the percentage of tau-PET scans which would be saved via blood-based screening. Confidence intervals around the number of saved tau-PET scans achieved while maintaining a certain sensitivity and P-values associated with comparisons were calculated using 100 bootstrap trials. PPVs at the derived cutoffs for each biomarker were also reported; this value represents the percentage of tau-PET positive individuals among those who would receive a tau-PET scan.

A sensitivity analysis was performed to calculate the number of tau-PET scans saved when looking at the SCD + MCI group and at the all-cause dementia group separately. The number of tau-PET scans saved in these groups was then compared to the results found in the entire study population. Also, a sensitivity analysis was performed by combining each plasma biomarker with age and *APOE* ε4 status in logistic regression models, and applying cutoffs according to the

sensitivities from screening strategies to each model's probability of tau-PET positivity output, and the number of saved scans was compared to that of the univariate blood biomarkers.

As a subsequent post-hoc analysis, we evaluated the added prognostic value of tau-PET over cognitive decline within the context of the proposed screening workflow, to assess whether tau-PET would indeed only be prognostically useful in the patients referred to a scan. For this analysis, the best-performing blood-based biomarker was chosen as the example screening biomarker based on the main analysis. First, the percent annual change in MMSE was calculated based on baseline and most recent study visit, varying from one to four years from baseline. Then, the study population was divided into those above the plasma pTau217 referral cutoff—representing those who would be referred to a tau-PET scan—and into those below the cutoff—those who would not be referred to a tau-PET scan. In each population separately, four regression models were fitted to investigate the added value of tau-PET in predicting change in MMSE, given the blood-based biomarker screening step: (i) a basic demographic model including age and sex as a reference; (ii) a model adding the continuous blood-based biomarker levels to the demographic model; (iii) a model adding tau-PET SUVr in the temporal meta-ROI to the demographic model; (iv) a model adding both blood-based biomarker and tau-PET SUVr to the demographic value.

To comparatively illustrate to what extent tau-PET and plasma pTau217 add value to prognostic assessment in each of these populations, we compared models with likelihood ratio tests, adjusted $R^2$ (a measure of how well a model explains outcome variability), and performed model averaging to yield the weighted Akaike Information Criterion (wAIC), a metric ranging from 0-1 indicating which model provides the best tradeoff between goodness-of-fit and model complexity[47–49]. These analyses were performed separately for the SCD + MCI (mean [SD] follow-up of $3 \pm 0.9$ years) and all-cause dementia (mean [SD] follow-up of $1.8 \pm 0.4$ years) groups.

All statistical analysis was performed using the R programming language (v4.0.0). When applicable, statistical tests were two-sided with an alpha level of 0.05.

### Reporting summary
Further information on research design is available in the Nature Portfolio Reporting Summary linked to this article.

## Data availability
Pseudonymized data will be shared by request from a qualified academic investigator for the sole purpose of replicating procedures and results presented in the article and as long as data transfer is in agreement with EU legislation on the general data protection regulation and decisions by the Swedish Ethical Review Authority and Region Skåne, which should be regulated in a material transfer agreement. Arrangements for data sharing for replication of the findings in the TRIAD data set are subject to standard data-sharing agreements, and further information can be found on the study's website (https://triad.tnl-mcgill.com/). Source Data File are provided for reproduction of key plots. Source data are provided with this paper.

## Code availability
The R code that supports the results of this study is publicly available on GitHub (https://github.com/wsbrum/bbm_taupet_npv). All models were built using publicly available packages and functions in the R programming language.

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

## Acknowledgements

The study was supported by the Swedish Research Council (2022-00775), ERA PerMed (ERAPERMED2021-184), the Knut and Alice Wallenberg Foundation (2017-0383), the Strategic Research Area MultiPark (Multidisciplinary Research in Parkinson's disease) at Lund University, the Swedish Alzheimer Foundation (AF-980907), the Swedish Brain Foundation (FO2021-0293), The Parkinson foundation of Sweden (1412/22), the Cure Alzheimer's fund, the Konung Gustaf V:s och Drottning Victorias Frimurarestiftelse, the Skåne University Hospital Foundation (2020-O000028), Regionalt Forskningsstöd (2022-1259) and the Swedish federal government under the ALF-agreement (2022-Projekt0080). WSB is supported by CAPES [88887.372371/2019-00] [88887.596742/2020-00] and Stiftelsen för Gamla Tjänarinnor. KB is supported by the Swedish Research Council (#2017-00915), the Alzheimer Drug Discovery Foundation (ADDF), USA (#RDAPB-201809-2016615), the Swedish Alzheimer Foundation (#AF-930351, #AF-939721 and #AF-968270), Hjärnfonden, Sweden (#FO2017-0243 and #ALZ2022-0006), the Swedish state under the agreement between the Swedish government and the County Councils, the ALF-agreement (#ALFGBG-715986 and #ALFGBG-965240), the European Union Joint Program for Neurodegenerative Disorders (JPND2019-466-236), the National Institute of Health (NIH), USA, (grant #1R01AG068398-01), and the Alzheimer's Association 2021 Zenith Award (ZEN-21-848495). N.M.C. is supported by the EU Joint Programme Neurodegenerative Diseases (2019-03401), the WASP and DDLS Joint call for research projects (WASP/DDLS22-066), the Swedish Brain Foundation (FO2023-0163), the Swedish Alzheimer Foundation (AF-994229), the Swedish Research Council (2021-02219) and the Swedish federal government under the ALF agreement (2022-Projekt0107). The TRIAD study is supported by the Weston Brain Institute, Canadian Institutes of Health Research (CIHR) [MOP-11-51-31; RFN 152985, 159815, 162303], Canadian Consortium of Neurodegeneration and Aging (CCNA; MOP-11-51-31 -team 1), the Alzheimer's Association [NIRG-12-92090, NIRP-12-259245], Brain Canada Foundation (CFI Project 34874; 33397), the Fonds de Recherche du Québec—Santé (FRQS; Chercheur Boursier, 2020-VICO-279314), CCNA Canadian Consortium of Neurodegeneration in Aging (theme 1, Team 2) and Colin J. Adair Charitable Foundation.

## Author contributions

W.S.B., N.C.C., N.M.C., and O.H. designed the study. W.S.B., S.J., A.L.B., H.Z., N.J.A., K.B., and O.H. acquired the blood biomarker data. J.T., S.J., N.R., J.S., S.S., A.L.B., E.R.Z., G.B.F., E.S., S.P., P.R.N., and O.H. provided technical support and acquired and processed neuroimaging and clinical data. W.S.B., N.C.C., and J.T. performed statistical analyses. W.S.B., N.C.C., N.M.C., and O.H. wrote the initial draft of the manuscript. All authors contributed to the interpretation of the results and to subsequent manuscript drafts.

## Funding

## Competing interests

J.S. has received research funding from Avid Radiopharmaceuticals (a wholly owned subsidary of Eli Lilly); consulted for Roche Pharmaceuticals, Biogen, Merck, and Eli Lilly; given educational lectures sponsored by GE Healthcare, Eli Lilly, and Biogen; and is Chief Medical Officer for ARUK. H.Z. has served at scientific advisory boards and/or as a consultant for Abbvie, Acumen, Alector, Alzinova, ALZPath, Amylyx, Annexon, Apellis, Artery Therapeutics, AZTherapies, Cognito Therapeutics, CogRx, Denali, Eisai, Merry Life, Nervgen, Novo Nordisk, Optoceutics, Passage Bio, Pinteon Therapeutics, Prothena, Red Abbey Labs, reMYND, Roche, Samumed, Siemens Healthineers, Triplet Therapeutics, and Wave; has given lectures in symposia sponsored by Alzecure, Biogen, Cellectricon, Fujirebio, Lilly, and Roche; and is a co-founder of Brain Biomarker Solutions in Gothenburg AB (BBS), which is a part of the GU Ventures Incubator Program (outside submitted work). N.J.A. has given lectures in symposia sponsored by Eli-Lilly, Roche Diagnostics, and Quanterix. N.J.A. has declined paid opportunities from ALZpath. K.B. has served as a consultant, on advisory boards, or at data monitoring committees for Abcam, Axon, BioArctic, Biogen, JOMDD/Shimadzu, Julius Clinical, Lilly, MagQu, Novartis, Ono Pharma, Pharmatrophix, Prothena, Roche Diagnostics, and Siemens Healthineers and is a cofounder of Brain Biomarker Solutions in Gothenburg AB (BBS), which is a part of the GU Ventures Incubator Program, outside the work presented in this paper. O.H. has acquired research support (for the institution) from ADx, AVID Radiopharmaceuticals, Biogen, Eli Lilly, Eisai, Fujirebio, GE Healthcare, Pfizer, and Roche. In the past 2 years, he has received consultancy/speaker fees from AC Immune, Amylyx, Alzpath, BioArctic, Biogen, Cerveau, Eisai, Eli Lilly, Fujirebio, Genentech, Merck, Novartis, Novo Nordisk, Roche, Sanofi, and Siemens. The remaining authors declare no competing interests.

## Additional information

[1]Department of Psychiatry and Neurochemistry, the Sahlgrenska Academy at the University of Gothenburg, Mölndal, Sweden. [2]Graduate Program in Biological Sciences: Biochemistry, Universidade Federal do Rio Grande do Sul (UFRGS), Porto Alegre, Brazil. [3]Clinical Memory Research Unit, Department of Clinical Sciences Malmö, Faculty of Medicine, Lund University, Lund, Sweden. [4]Wallenberg Center for Molecular Medicine, Lund University, Lund, Sweden. [5]McGill Centre for Studies in Aging, McGill University, Verdun, Quebec, QC, Canada. [6]Department of Neurology and Neurosurgery, Faculty of Medicine, McGill University, Quebec, QC, Canada. [7]Department of Pharmacology, Universidade Federal do Rio Grande do Sul (UFRGS), Porto Alegre, Brazil. [8]Graduate Program in Biological Sciences: Pharmacology, Universidade Federal do Rio Grande do Sul (UFRGS), Porto Alegre, Brazil. [9]Memory Clinic, Skåne University Hospital, Lund, Sweden. [10]Clinical Neurochemistry Laboratory, Sahlgrenska University Hospital, Mölndal, Sweden. [11]Department of Neurodegenerative Disease, UCL Institute of Neurology, Queen Square, London, UK. [12]UK Dementia Research Institute at UCL, London, United Kingdom. [13]Hong Kong Center for Neurodegenerative Diseases, Hong Kong, China. [14]Wisconsin Alzheimer's Disease Research Center, School of Medicine and Public Health, University of Wisconsin–Madison, Madison, WI, USA. [15]Memory Center, Geneva University and University Hospital, Geneva, Switzerland. [16]King's College London, Institute of Psychiatry, Psychology and Neuroscience Maurice Wohl Institute Clinical Neuroscience Institute, London, UK. [17]NIHR Biomedical Research Centre for Mental Health and Biomedical Research Unit for Dementia at South London and Maudsley NHS Foundation, London, UK. [18]Centre for Age-Related Medicine, Stavanger University Hospital, Stavanger, Norway. [19]Department of Neurology, Skåne University Hospital, Lund, Sweden. [20]These authors contributed equally: Wagner S. Brum, Nicholas C. Cullen, Joseph Therriault. ✉e-mail: oskar.hansson@med.lu.se

