## [Peer Review File · Nature Communications]

A blood-based biomarker workflow for optimal tau-PET referral in memory clinic settingsEditorial Note: Parts of this Peer Review File have been redacted as indicated to remove third-party material where no permission to publish could be obtained.

REVIEWER COMMENTS

Reviewer #1 (Remarks to the Author):

Doesn't the closer correlation between plasma p-tau markers and Ab-PET give one pause as to the utility of this marker to filter down those who need a tau-PET scan specifically?

Did you consider scaling these sensitivity estimates by age, sex and the cases where the patient may have known APOE4 information?

While BioFINDER is an interesting cohort, it is complicated in some ways - it is a smaller dataset when considered alone, and has a generally much younger profile for patients than would be realistic in a clinical setting. It is also a European cohort and there might be some differences with the United States. Did the authors consider other memory clinical datasets that are available in the US? Or even in the ADNI dataset as a comparison?

I'm not entirely understanding the scientific/theoretical component to this work - I wonder about the fit of this work for this type of journal? Perhaps this might be more suited to a directly clinical neurology audience? Or a clinical trialist audience?

Did the authors compare different platforms? This is using MS/immunoassay and various Simoa platforms, but how might this differ for other platforms? There is considerable variability here, as well as marked considerations for what is most likely available at a commercial level (i.e., in a clinic).

In addition, tau-PET is measured using RO948, but this is not the Tauvid tracer (Flortaucipir) that will be used in the clinic, which is a huge drawback. Right now, it seems that there is an oranges to apples comparison.

In addition, do the authors think that this temporal meta-ROI and the corresponding cut-off approach a likely use of the tau-PET screening in-clinic? Wouldn't it be more appropriate to be using visual detection (visual read) rather than a quantification method that is less likely to be used in-clinic?

What is the level of diversity in this cohort? Ethnicity will have a profound impact on generalizability of these findings on the general memory clinic population, particularly in the United States (depending on the state that is being considered).

The authors need to define and describe what the weighted AIC is in the results section rather than leaving only in the methods. This is helpful to know so the reader can understand why they need to know this number and how to interpret it. Is a higher number better in this case? Does a poorer fitting model get penalized to 0? And what is considered to be a suitable wAIC threshold for these models in this case?

Wouldn't a more interpretable model be the AUC to predict a threshold of cognitive decline? Given that cognitive decline is rather heterogeneous, just looking at model fit doesn't give you much in the way of

clinical interpretation.

Reviewer #2 (Remarks to the Author):

Brum, Cullen, et al. investigate whether the number of uninformative Tau PETs could be reduced by screening with blood-based biomarkers patients with cognitive complaints attending a Memory clinic. The authors took the a priori decision that blood-based biomarkers should be used to screen-out patients with a low likelihood of having a positive Tau PET, but not exclude patients with a reasonable likelihood of being Tau PET positive. Thus, sensitivity was favour over specificity. They calculated the cut-offs of the main plasma biomarkers, namely p-tau181, p-tau217, p-tau231, GFAP, A β 42/40 and NfL corresponding to a Sensitivity of 90%, 95% and 99%. With these cut-offs, they calculated the number of Tau PET scans saved if that PET was not performed on those patients below that cut-off. They also calculated the positive predictive value and, finally, the prognostic value (as measured by the association with follow-up MMSE). In all these analyses, plasma p-tau217 showed the best performance, while plasma A β 42/40 and NfL showed the worst performance. The authors propose a two-step decision tree, based on plasma p-tau217 screening, to optimise the selection of patients that should undergo Tau PET.

This is an interesting study that have clinical implications in patient management. I would recommend the acceptance in Nature Communications after some points are addressed:

- The positive predictive value is affected by the prevalence. In fact, the PPV the authors calculate for each biomarker is considerably lower in the SCD-MCI group (Suppl Fig 1) compared to the whole sample (Figure 2) or the dementia group (Suppl Fig 2). The authors discuss that is out of the scope of this study simulating different prevalences, which I understand. Still, I believe that they should discuss whether the screening workflow they propose to optimise tau PET would be equally useful in patients with SCD-MCI and those with dementia.
- MMSE is not the best cognitive test to study the prognostic value of a biomarkers, are there other cognitive tests available?
- Plasma A β 42/40, measured with Araclon assay, did not render the best performance. However, the authors should mention that other A β 42/40 assays may lead to better results. Not to mention that, if they have other plasma A β 42/40 measurements, they could consider it including them.
- Please indicate whether the diagnoses of "AD" and "non-AD" (Table 1) are clinical diagnoses (and criteria used) or are biomarker-based.
- The authors use indistinctly in the text pre-screening or screening. I would suggest to be consistent.

Reviewer #3 (Remarks to the Author):

In this manuscript, the authors delve into the possibility of using plasma for pre-screening tau PET referrals. Similar to previously published studies from this group, the authors use the same BioFinder data to show that pTau217 is more sensitive than pTau231, pTau181, NFL, GFAP, and Ab42/Ab40. They report that screening with pTau217 would “save” 45% of tau PET scans (i.e., these scans would be unnecessary). They also show that there is a 71% tau-PET-positivity rate among those who would be referred for a scan. While the idea of pre-screening for tau PET scans is appealing, I believe that the recommendations proposed by the authors need to be tempered.

Because the authors defined tau-PET-positivity using a bilateral temporal meta-ROI, the claims about tau-PET positivity can only be applied to determining whether pTau217 can effectively screen for tau burden within this particular region. The regional distribution of tau can vary greatly and while the majority of individuals may follow Braak staging, there have been many reported cases of low temporal tau but high extra-temporal tau. Indeed, the authors show that pTau217 has a PPV of 71%, but this also means that 29% of those who were pTau217-positive were tau-PET-negative. What is driving this discrepancy? Could these be individuals who show atypical tau-PET patterns? Additionally, are there individuals who are pTau217-negative but have significant tau PET signal outside of the temporal meta-ROI? Based on the results provided in this study, it is unknown if pTau217 will correctly screen those with extra-temporal tau. If pTau217 does not accurately screen-in those with atypical tau PET patterns, the authors proposal of a two-step workflow for clinical work-up will actually exacerbate biases towards a more homogeneous definition of AD that is focused on temporal tau.

Related to this concern, the authors show that 10% of their AD group are tau-PET-negative. Do these 10% represent misdiagnoses or do these individuals have atypical tau PET patterns that are not captured with their temporal meta-ROI?

Another conceptual concern I have is that given that some studies have suggested that pTau changes occur earlier than tau-PET changes, is it more beneficial to use pTau to screen out those who are not tau-PET positive, rather than using it as a screen-in tool? In a clinical context, the benefit of confirming tau involvement using plasma pTau and then tau-PET is unclear to me (though of course it’s interesting from a research and clinical trial perspective, but the title says “in memory clinic settings”), whereas I can imagine it would be helpful to rule out tau involvement for a clinical patient.

An important detail that is missing is the time lag between the blood draw and the tau-PET scans.

Cognition here was only measured with the MMSE, which is not a comprehensive assessment of cognitive ability. This is an important limitation to consider.

A minor point – I think at least one of the references is incorrect. Reference 24 is listed as the reference for the temporal meta-ROI definition but Cullen et al. does not include PET data.

Reply to the reviewers' comments:

We believe these additions further support the robustness of our findings. Please find below a detailed response to each reviewer in blue, with significant changes and additions made in the revised article document also marked.

Reviewer #1:

Remark #1: Doesn't the closer correlation between plasma pTau markers and Ab-PET give one pause as to the utility of this marker to filter down those who need a tau-PET scan specifically?

Response: We thank the reviewer for carefully reading our manuscript and for this question. Plasma pTau is associated with both A β and tau pathologies, when the latter are measured with A β -PET, tau-PET or immunohistochemistry post-mortem (Mattsson et al 2021 PMID: 33949133; Therriault et al 2023 PMID: 36508198; Salvadó et al 2023 PMID: 36912178). In fact, it seems that the highest increases in plasma pTau217 occur in individuals with more advanced stages of tau pathology (Palmqvist et al PMID: 32722745). Additionally, the greater associations between plasma p-tau and A β -PET are more prominent in asymptomatic/preclinical individuals (Therriault et al 2023 PMID: 36508198), which is not the population of the current proposal.

Below is Figure 2 of the mentioned paper by Salvadó et al, showing that while plasma A β 42/A β 40 is mostly associated with plaque pathology, plasma pTau217 is substantially associated with plaque and tangle pathology, with even higher associations between pTau217 and tau tangles:

[REDACTED]

Thus, we understand that pTau biomarkers can be a feasible option to be used in a tau-PET scan referral workflow like this one, especially considering the results achieved in this work.

Remark #2: Did you consider scaling these sensitivity estimates by age, sex and the cases where the patient may have known APOE4 information?

Response: We thank the reviewer for this suggestion. We agree that doing analyses taking APOEε4 carriership is important, given its strong association with AD. We think, however, that a more practical alternative is to incorporate these suggested predictors (APOEε4 status, age and sex) into a risk prediction model and evaluate the screening strategy with probability cutoffs based on the adjust model probability output, rather than simply deriving several different cutoffs for each subgroup of interest. This is the same approach done, for example, by WashU/C2N with their amyloid probability score (Hu et al, JAMA Net Open PMID) and in our recent two-step workflow for Aβ-positivity (Brum et al, Nat Aging, PMID 37653254). The probability incorporates the adjustments related to these variables and then a single probability cutoff can be used for all.

We have conducted an analysis in both BioFINDER-2 and TRIAD comparing how many scans could be further saved by adding age, APOE and sex information into the screening strategy through the use of probability cutoffs. This has now been added in the Supplement (Supp. Figure 1) and in the main text Results and Methods.

Supp. Figure 1.: Combining age and APOE ε4 status to individual biomarkers did not lead to increases in saved scans.

Methods (Page 19 Line 18): “Also, a sensitivity analysis was performed by combining each plasma biomarker with age and APOE ε4 status in logistic regression models, and applying cutoffs according to the sensitivities from screening strategies to each model’s probability of tau-PET positivity output, and the number of saved scans was compared to that of the univariate blood biomarkers.”

Results (Page 6 Line 10): “In a sensitivity analysis, we evaluated whether adding age, sex and APOE ε4 status to each plasma biomarker would increase the percentage of tau-PET scans saved. In both TRIAD and BioFINDER-2, this generally led to slight increases in saved scans for all biomarkers, but none of these increases were significant (Supp. Figure 1).”

Remark #3: While BioFINDER is an interesting cohort, it is complicated in some ways - it is a smaller dataset when considered alone, and has a generally much younger profile for patients than would be realistic in a clinical setting. It is also a European cohort and there might be some differences with the United States. Did the authors consider other memory clinical datasets that are available in the US? Or even in the ADNI dataset as a comparison?

Response: We thank the reviewer for this comment.

We agree that adding a new cohort would increase the generalizability of the results. We have therefore added results from TRIAD, a cohort based in Montreal, Canada (McGill University). The results were very similar as shown by the new figures 1 and 2 (inserted below) and detailed in the results. This shows that the results are not specific to the Swedish BioFINDER study. Further, in TRIAD a different pTau217 assay was used (Janssen Simoa assay), and a different tau-PET tracer (^{18}F JK6240) also showing that the results are not specific for a certain pTau217 assay or a certain tau-PET tracer.

Regarding the age of the participants. The mean age in the BioFINDER-2 participants included here was 70.9 years with a range from 51 to 93 years. In many countries the oldest old are often managed in primary care (and not specialist memory clinics) even though this can vary substantially by the resources and priorities of the different healthcare systems. Further, the included population is not substantially different in age when comparing to many other aging and dementia cohorts across the globe, as well as cross-sectional memory clinic case series. In a recent head-to-head comparison of pTau biomarkers within the BioDEGMAR cohort, the mean (SD) age of a cognitively impaired, real-world memory clinic-based population (n=197) was of 72.3 (5.8) years (Ashton et al, PMID 36370462). In the Paris Laboisier memory clinic-based cohort, the mean (SD) age for MCI A β - and MCI A β + was 66.3 (9.6) and 72.4 (7.9), respectively (Lantero-Rodriguez et al, PMID 36899441). In the AIBL study, the mean (SD) age for MCI A β - (n=26) and MCI A β + (n=33) was 71.3 (11.5) and 75.6 (5.7), respectively, in a previous study (Chatterjee et al, PMID 36574591). In the Harvard Aging Brain Study (HABS), MCI participants (n=51) showed a mean (SD) age of 72.8 (8.8) years (Hanseeuw et al, PMID 29792874). In summary, the BioFINDER-2 patient age is in line with other research or memory clinic-based cohorts, as well as for the newly included TRIAD participants, with mean age of 70.0 (7.7) years.

Figure 1: Tau PET scans saved as a function of biomarker sensitivity.

Results (Page 5 Line 15) regarding Figure 1: “As expected, there was a tradeoff between the percentage of total tau-PET scans saved and the sensitivity for tau-PET positivity (Fig. 1A). At 95% sensitivity (i.e., theoretically missing 5% of positive tau-PET scans), the percentage of tau-PET scans saved ranged from 11.1–42.3% for different plasma biomarkers in BioFINDER-2 and from 5.4–44.6% in TRIAD. Notably, plasma pTau217 screening led to significantly higher percentages of reduced tau-PET scans (BioFINDER-2: 42.3%; TRIAD: 44.6%) compared with all other biomarkers ($P < 0.001$ for all comparisons in both cohorts), followed by GFAP (BioFINDER-2: 22.5%; TRIAD: 20.2%), pTau181 (BioFINDER-2: 19.4%; TRIAD: 16.7%), pTau231 (BioFINDER-2: 15.3%; TRIAD: 16.7%), Aβ42/Aβ40 (BioFINDER-2: 14.8%; TRIAD: 5.9%), and NfL (BioFINDER-2: 11.1%; TRIAD: 5.4%) (Fig. 1B). A similar ranking of blood-based biomarkers was observed when evaluating a more lenient screening strategy with sensitivity at 90% or a more stringent strategy with 97.5% sensitivity. At 90% sensitivity (i.e., missing 10% of tau-PET-positive patients), screening with pTau217 saved 50.7% and 52.1% of tau-PET scans in BioFINDER-2 and TRIAD, respectively. At 97.5% sensitivity, the percentage reduction in scans achieved by pTau217 screening was

33.1 and 41.6% in BioFINDER-2 and TRIAD, respectively. At 90% sensitivity, other biomarkers saved 15.8-31.9% (BioFINDER-2) and 8.1-52.1% (TRIAD) of tau-PET scans with GFAP and pTau181 following pTau217 (31.9 and 29.5% saved scans). At 97.5% sensitivity, other biomarkers' ability to save tau-PET scan ranged from 6.5-11.8% (BioFINDER-2) and from 5.9-13.6% (TRIAD).”

Figure 2. Positive predictive value at different sensitivity thresholds.

Results (Page 7 Line 1): “Next, positive predictive value (PPV) was calculated as the percentage of tau-PET-positive individuals among those who would be selected for tau-PET according to their plasma biomarker results (Fig. 2). In both BioFINDER-2 and in TRIAD, the only plasma biomarker to show a PPV significantly ($p < 0.05$) greater than the study population rate of tau-PET positivity (BioFINDER-2: 37.0%; TRIAD: 44.1%) at all evaluated strategies (90, 95 and 97.5% sensitivity) was pTau217. At 95% sensitivity, the PPVs ranged from a 1.8–30.3 percentage point increase over the population rate in BioFINDER-2 and from 3.1-33.1% in TRIAD (except for Aβ42/Aβ40 which led to -4.0%). These increases over the population rate at this sensitivity (95%) were only significant for pTau217 and GFAP in BioFINDER-2 and only for pTau217 in TRIAD. At 95% sensitivity, pTau217 significantly outperformed all other biomarkers ($p < 0.0001$), giving a PPV of 67.3% (95% CI 52.6-77.0%) in BioFINDER-2 and of 77.1% (95% CI 60.1-94.6%) in TRIAD. In both cohorts, similar results favoring pTau217 were seen for screening cutoffs at 90%, with PPVs of 73.1% in BioFINDER-2 and of 86.2% in TRIAD, or when using screening cutoffs at 97.5% sensitivity, with PPVs of 60.1% in BioFINDER-2 and 73.1% in TRIAD (Fig. 2).”

Remark #4: I'm not entirely understanding the scientific/theoretical component to this work - I wonder about the fit of this work for this type of journal? Perhaps this might be more suited to a directly clinical neurology audience? Or a clinical trialist audience?

Response: We thank the reviewer for this comment. This paper is under review within the "Clinical Research" series, and this journal has previously published several clinical papers within our field.

Remark #5: Did the authors compare different platforms? This is using MS/immunoassay and various Simoa platforms, but how might this differ for other platforms? There is considerable variability here, as well as marked considerations for what is most likely available at a commercial level (i.e., in a clinic).

Response: We thank the reviewer for this comment. We mostly used Simoa immunoassays, but also two Meso Scale Discovery (MSD) immunoassays. Further, a mass-spectrometry assay was used to quantify plasma A β in BioFINDER-2. It is not yet clear what will be the specific assays or platforms that will be widely established for global use in the future, but, in general, immunoassays are easier to implement in local hospitals. For pTau217, we here used the Lilly-developed assay on the MSD platform in BioFINDER-2 and the Janssen-developed assay on the Simoa platform in TRIAD with remarkably similar results. As far as we understand, the Janssen assay is being commercialized, and very likely also the Lilly assay.

Remark #6: In addition, tau-PET is measured using RO948, but this is not the Tauvid tracer (Flortaucipir) that will be used in the clinic, which is a huge drawback. Right now, it seems that there is an oranges to apples comparison.

Response: We humbly disagree with the reviewer on this point. The molecular structures of RO948 and FTP are extremely similar (only differ by one atom). Not surprisingly, we found in a head-to-head study that the neocortical SUVR tracers correlate linearly between the two tracers giving virtually the same SUVR values (Smith et al, 2020; PMID: 31612245). Similarly, we found that the cut-offs used to define an abnormal tau-PET scan are virtually the same when using these two tracers (Leuzy et al, 2021; PMID: 34041562). The latter paper also showed very similar cut-offs for abnormality of MK6240 scans. MK6240 was also the tracer used in TRIAD, with very similar results to those obtained in BioFINDER (using RO948) indicating that the results are not limited to a certain tau-PET tracer.

Further, RO948 has been used to determine PET positivity in studies merging data from different cohorts, including those using Flortaucipir. Most notably, a study by Ossenkopele et al (2022, Nat Med, PMID: 36357681) showed that tau positivity (defined either by RO948 or flortaucipir) is strongly associated with future cognitive decline even in cognitively unimpaired participants, further supporting that, despite individual tracer particularities, they are both effective in the prognostication according to tau positivity.

Additionally, during the revisions of our present work, our group has published what we consider to be an important paper for the clinical implementation of tau PET (Smith et al, JAMA Neuro, 2023, PMID: 37213093). We used RO948 and validated a *visual read* method for this ligand within BioFINDER-2 participants. BioFINDER-2 is a memory clinic-based study, which enables the evaluation of research-based tools into patient care and their evaluation. This work showed that a visual read of tau-PET scan significantly improved the clinical management of patients who had already undergone a very thorough clinical and biomarker characterization.

Remark #7: In addition, do the authors think that this temporal meta-ROI and the corresponding cut-off approach a likely use of the tau-PET screening in-clinic? Wouldn't it be more appropriate to be using visual detection (visual read) rather than a quantification method that is less likely to be used in-clinic?

Response: We thank the reviewer for this comment. Following our previous response, we do agree that a visual read method can be clinically valuable. To further increase the clinical value of our current manuscript, we have added secondary analyses in the supplement (Supp. Fig. 2-3; Supp. Table 3), demonstrating that very similar results would be obtained had we used *visual read* of RO948 within the included BioFINDER-2 participants. These new results show very similar results to those obtained using quantitative assessments of the scans. This analysis has not been performed for the newly included TRIAD data considering a visual read method for MK6240 has not yet been validated.

Supplementary Figure 2: Defining tau-PET positivity with a validated visual read method leads to similar results in terms of avoided scans.

Results regarding saved scans (Page 6 Line 14): “*In BioFINDER-2, we also conducted a sensitivity analysis in which the same univariate screening strategies were evaluated but with the outcome as a positive tau-PET scan determined by a novel clinically validated method for visual interpretation of the RO1498 tau tracer.⁶ The results were generally similar, with pTau217 performing best, followed by GFAP and pTau181 in a second tier, with pTau231, Aβ42/Aβ40 and NfL performing worse in terms of saved scans (Supp. Figure 2). For example, screening for a visually positive tau-PET scan with plasma pTau217 reducing 50.2% scans at 90% sensitivity, 41.6% at 95% sensitivity and saving 30.0% of scans at 97.5% sensitivity.*”

Supplementary Figure 3. Defining tau-PET positivity with a validated visual read method leads to similar results in terms of positive predictive values.

Results regarding PPV's (Page 7 Line 16): *“When repeating the PPV analysis using tau-PET positivity determined with the visual read method in BioFINDER-2, pTau217 was again the only biomarker to lead to significant increases over the population rate of visual tau-PET positivity (45.3%). At 95% sensitivity, pTau217, showed a very similar PPV (66.5%, 95% CI 53.9-77.0%; 21.2% increase over population rate) as that of the quantitative-cutoff analysis above. At 90% and 97.5% sensitivities, pTau217 showed PPV's of 74.0% (95% CI 63.5-81.9%) and 57.3% (95% CI 46.2-70.9%), respectively.”*

Remark #8: What is the level of diversity in this cohort? Ethnicity will have a profound impact on generalizability of these findings on the general memory clinic population, particularly in the United States (depending on the state that is being considered).

Response: We thank the reviewer for this comment. The level of diversity in the BioFINDER-2 cohort is generally consistent with the population referred to our memory clinic in Malmö from the Swedish primary care system.

As a general outlook on this process, details on the referral origin for BioFINDER-1 (cohort not included here, but has a generally similar framework as BioFINDER-2), for which enrollment is completed, has been previously published (Materials and Methods section in Petrazzuoli et al, 2020, PMID 32417771): “The patients were mostly referred from primary care (80.8%), but 12.5% of the referrals came from other

specialist clinics and 6.7% were self-referrals”. For BioFINDER-2, for which the enrollment is still ongoing, such analyses have not yet been published. We recently looked at our records to characterize this for a subset of MCI participants from BioFINDER-2, and figures looked similar: most were referred from primary care (n=179; 84.4%), followed by hospital referrals (n=31; 14.6%) and self-referrals (n=2; 0.9%).

This is all to support that, despite being a research cohort, most of the referrals indeed come from primary care, which leads to a fairly good representativeness within what can be achieved based on southern Sweden’s demographic constitution. Additionally, the mean educational attainment for these participants was 12.4 years, which is substantially lower than most North American cohorts, including that of the TRIAD participants now included in this work (mean educational attainment of 15.1 years).

For BioFINDER-2, self-reported race/ethnicity has not been collected as part of the study protocol. Among the n=179 TRIAD participants, n=3 (1.7%) were Asian, n=1 was black (0.5%), n=172 were white (96%) and unknown/not reported for 3 (1.7%).

A consideration regarding these results and further need for evaluation of these biomarkers in under-represented populations has been made (Page 15 Line 8): *“While race/ethnicity was not available for BioFINDER-2, among the 179 included TRIAD participants, 3 (1.7%) were Asian, 1 was black (0.5%), n=172 were white (96%) and unknown/not reported for 3 (1.7%). Thus, further studies in more diverse populations are warranted.”*

We have also emphasized that many participants in the BioFINDER-2 study present with frequent age-related comorbidities, which reflects what is observed in general in populations of older adults (Page 15 Line 5): *“Additionally, many included patients from BioFINDER-2 present age-related comorbidities as observed in the real-world populations of older adults, with 50.4% (n=277) presenting with cardiovascular disease, 14.4% (n=79) with diabetes and 34.4% (n=189) with dyslipidemia.”*

Remark #9: The authors need to define and describe what the weighted AIC is in the results section rather than leaving only in the methods. This is helpful to know so the reader can understand why they need to know this number and how to interpret it. Is a higher number better in this case? Does a poorer fitting model get penalized to 0? And what is considered to be a suitable wAIC threshold for these models in this case?

Response: We appreciate the reviewer’s comment to improve our reporting. Indeed, a higher number is better. Among a set of nested models (in our case, for each data subset, e.g. the four models within plasma

pTau217 negative individuals with all-cause dementia), the wAIC of the models will add up to 1, and each it can be interpreted as a probability that, among these models, the one with the highest wAIC has the higher probability to provide the best trade-off between parsimony and goodness-of-fit.

We have now added a line on this in the results (Page 9 Line 16): “*We compared models using R^2 , which indicates how well the model explains outcome variability (the higher the better), and using weighted Akaike Information Criteria (wAIC) a metric ranging from 0-1 indicating which model, among a set of compared nested models, provides the best tradeoff between goodness-of-fit and model complexity (wAIC of compared models sums to 1, and the higher wAIC, the higher probability that the model provides the best trade-off).*”

Remark #10: Wouldn't a more interpretable model be the AUC to predict a threshold of cognitive decline? Given that cognitive decline is rather heterogeneous, just looking at model fit doesn't give you much in the way of clinical interpretation.

Response: We thank the reviewer for this important comment on clinical utility of the prognostic session of our paper. We have opted to keep R^2 and wAIC as we understand that they provide more clear information on how well the models explain the outcome variability (i.e. how well they explain cognitive decline), since they are based on a continuous metric (change in longitudinal cognitive scores). Using a binary cognitive outcome (i.e. clinical progression: yes or no), which is what would enable the use of the suggested AUC metric, is an approach that, compared with a continuous outcome (like the one we used) leads to substantial loss of statistical power, and we have not done so given this prognostic analysis is already stratified into several subgroups. Also, the point of the prognostic analyses was just to demonstrate that a tau PET scan will be prognostically informative mostly in those above the referral cutoff, but not in those screened-out (that would not undergo a scan).

Reviewer #2:

Brum, Cullen, et al. investigate whether the number of uninformative Tau PETs could be reduced by screening with blood-based biomarkers patients with cognitive complaints attending a Memory clinic. The authors took the a priori decision that blood-based biomarkers should be used to screen-out patients with a low likelihood of having a positive Tau PET, but not exclude patients with a reasonable likelihood of being Tau PET positive. Thus, sensitivity was favour over specificity. They calculated the cut-offs of the main plasma biomarkers, namely pTau181, pTau217, pTau231, GFAP, A β 42/40 and NfL corresponding to a Sensitivity of 90%, 95% and 99%. With these cut-offs, they calculated the number of Tau PET scans saved if that PET was not performed on those patients below that cut-off. They also calculated the positive predictive value and, finally, the prognostic value (as measured by the association with follow-up MMSE). In all these analyses, plasma pTau217 showed the best performance, while plasma A β 42/40 and NfL showed the worst performance. The authors propose a two-step decision tree, based on plasma pTau217 screening, to optimise the selection of patients that should undergo Tau PET.

This is an interesting study that have clinical implications in patient management. I would recommend the acceptance in Nature Communications after some points are addressed:

Remark #1: The positive predictive value is affected by the prevalence. In fact, the PPV the authors calculate for each biomarker is considerably lower in the SCD-MCI group (Suppl Fig 1) compared to the whole sample (Figure 2) or the dementia group (Suppl Fig 2). The authors discuss that is out of the scope of this study simulating different prevalences, which I understand. Still, I believe that they should discuss whether the screening workflow they propose to optimise tau PET would be equally useful in patients with SCD-MCI and those with dementia.

Response: We thank the reviewer for taking the time to carefully read our manuscript and for the positive comments. Indeed, we agree that discussing this is important.

We have added the following consideration in the Discussion (Page 14 Line 22): *“While our screening process showed good results even in the SCD-MCI subgroup analyses (a population with lower prevalence of tau-PET positivity and thus with lower PPVs for the studied diagnostic tests), the results were substantially better for the all-cause dementia population, which is compatible with the well-described fact that current AD tau biomarkers perform better in advanced stages”*.

Remark #2: MMSE is not the best cognitive test to study the prognostic value of a biomarkers, are there other cognitive tests available?

Response: We thank the reviewer for this comment. Indeed, while MMSE might not be the most sensitive cognitive test, it has generally worked well on a group-level in observational studies including symptomatic patients, such as in BioFINDER (Mattsson-Carlgrén et al, JAMA Neuro 2023, PMID: 36745413; Leuzy et al, JAMA Neuro 2021, PMID: 34928318) and ADNI (Donohue et al, JAMA 2017, PMID: 28609533). The main point of the prognostic analyses was simply to show that the proposed referral workflow could effectively screen-in and screen-out individuals in a way that was also prognostically meaningful (which was shown), rather than to provide a clinical tool to predict decline in new patients. Thus, we opted to preserve change in MMSE as the main outcome.

We attach a plot for the reviewer with the raw trajectories for each subpopulation used in our prognostic analyses so it can be seen that, in this subset of patients, there is a clear cognitive decline signal captured by MMSE:

Remark #3: Plasma A β 42/40, measured with Araclon assay, did not render the best performance. However, the authors should mention that other A β 42/40 assays may lead to better results. Not to mention that, if they have other plasma A β 42/40 measurements, they could consider it including them.

Response: We thank the reviewer for this comment.

Indeed, plasma A β 42/A β 40 assays present a considerable heterogeneity in their performance to detect confirmed A β pathology. Importantly, even the assays regarded as “best-performing”, such as the WashU/C2N mass-spec assay, present heterogeneity in their performance, recently presenting AUCs of 0.76 in a secondary analysis of A4 participants (Winston et al, JAD, 2023, PMID: 36710683). As previously described and discussed in several papers, the issue with plasma A β is more biological than assay-based, as the ratio is only reduced by 8-14% in the plasma of AD patients (Janelidze et al, JAMA Neuro, 2021, PMID: 34542571), which creates a robustness issue for the biomarker in plasma (Benedet et al, 2022, PMID: 35130933; Rabe et al, 2023, PMID: 36150024). That said, the Araclon assay is the only one available in large numbers for BioFINDER-2.

Nevertheless, it is known that biofluid measures of A β are poorly associated with measures of tau pathology (e.g. Figure 2 of Salvadó et al 2023 PMID: 36912178), so results wouldn't likely be altered given that in this work the focus was on screening for tau-PET abnormalities.

We have now added a remark on this in the Discussion (Page 14 Line 16): “*The plasma A β assay available for BioFINDER-2 was the Araclon assay, which has not shown the best performance in recent head-to-head comparisons. However, A β in the plasma presents a well-described robustness issue^{1,2} and it's not substantially associated with tau pathology, so results would not likely change substantially had we used a different A β assay*”.

Remark #4: Please indicate whether the diagnoses of “AD” and “non-AD” (Table 1) are clinical diagnoses (and criteria used) or are biomarker-based.

Response: We thank the reviewer for this comment which can help us to improve the clinical description of the cohort.

All patients underwent thorough clinical and neuropsychological assessments. A diagnosis of MCI or SCD was clinical. For AD, patients needed to fulfill the DSM-5 criteria for major neurocognitive disorder due to AD. In BioFINDER-2, CSF A β 42/A β 40 was assessed by the experts providing the diagnosis, according to

the NIA-AA recommendations, and in very few cases in the dataset herein included patients were given an AD diagnosis (n=3 CSF A β -negative AD patients). This was done clinically on an individual-case basis where there was either other evidence of A β pathology (e.g. PET) or borderline CSF cases. In TRIAD, the same process was performed, with only a few A β -negative AD cases (n=7). For the several non-AD disorders, diagnoses were clinically based (without any A β -status influence on the final diagnostic) according to the following criteria: DSM-V for FTD and vascular dementia; Gelb et al for PD (PMID: 9923759); Höglinger et al for PSP (PMID 28467028); Gilman et al for MSA (PMID: 18725502); Armstrong et al for CBS (PMID: 23359374); Gorno-Tempini et al for PPA (PMID: 21325651).

We have added a brief note on this information in the methods section (Page 17 Line 6): *“Among participants referred to the memory clinic not meeting criteria for dementia as per DSM-5, those performing worse than -1.5 SD in any cognitive domain, based on age and education norms, were classified as having MCI. Those not meeting MCI criteria were classified as having SCD. For AD dementia, participants needed to meet DSM-5 criteria for dementia due to AD (major neurocognitive disorder) and present biomarker-evidence of A β -positivity as per the NIA-AA guidelines, with exceptions to this (e.g. borderline cut-off patients) being evaluated in an individual-case basis. Patients without typical AD clinical syndromes had a clinical diagnosis of frontotemporal dementia, vascular dementia, Parkinson’s disease, progressive supranuclear palsy, multiple systems atrophy, corticobasal syndrome or primary progressive aphasia made based on appropriate clinical guidelines for each disorder”*.

Remark #5: The authors use indistinctly in the text pre-screening or screening. I would suggest to be consistent.

Response: We thank the reviewer for spotting this inconsistency. All occurrences now have been changed to “screening”.

Reviewer #3:

In this manuscript, the authors delve into the possibility of using plasma for pre-screening tau PET referrals. Similar to previously published studies from this group, the authors use the same BioFinder data to show that pTau217 is more sensitive than pTau231, pTau181, NFL, GFAP, and Ab42/Ab40. They report that screening with pTau217 would “save” 45% of tau PET scans (i.e., these scans would be unnecessary). They also show that there is a 71% tau-PET-positivity rate among those who would be referred for a scan. While the idea of pre-screening for tau PET scans is appealing, I believe that the recommendations proposed by the authors need to be tempered.

Remark #1: Because the authors defined tau-PET-positivity using a bilateral temporal meta-ROI, the claims about tau-PET positivity can only be applied to determining whether pTau217 can effectively screen for tau burden within this particular region. The regional distribution of tau can vary greatly and while the majority of individuals may follow Braak staging, there have been many reported cases of low temporal tau but high extra-temporal tau. Indeed, the authors show that pTau217 has a PPV of 71%, but this also means that 29% of those who were pTau217-positive were tau-PET-negative. What is driving this discrepancy? Could these be individuals who show atypical tau-PET patterns? Additionally, are there individuals who are pTau217-negative but have significant tau PET signal outside of the temporal meta-ROI? Based on the results provided in this study, it is unknown if pTau217 will correctly screen those with extra-temporal tau. If pTau217 does not accurately screen-in those with atypical tau PET patterns, the authors proposal of a two-step workflow for clinical work-up will actually exacerbate biases towards a more homogeneous definition of AD that is focused on temporal tau.

Response: We thank the reviewer for finding the time to carefully read our manuscript and for several constructive comments.

Indeed, as acknowledged in the limitations of the originally submitted version, dichotomization of outcomes is an ever-present issue in clinical papers from all areas, since some threshold for decision-making is always necessary. This becomes important to discuss for tau PET, given the heterogeneity in tau distribution underlying different manifestations of AD.

We have now included secondary analyses in which tau-PET has been clinically interpreted with a novel visual reading method, in which participants are considered to present a negative, intermediate or advanced burden, which overcomes the limitations associated with a meta-temporal ROI-based cutoff for tau-

positivity. Indeed, results were very similar between defining tau-PET positivity quantitatively or with the visual method (Supp. Figures 2 and 3; Supp. Table 3).

This has been incorporated in the results section:

Results regarding saved scans (Page 6 Line 14): *“In BioFINDER-2, we also conducted a sensitivity analysis in which the same univariate screening strategies were evaluated but with the outcome as a positive tau-PET scan determined by a novel clinically validated method for visual interpretation of the RO1498 tau tracer.1 The results were generally similar, with pTau217 performing best, followed by GFAP and pTau181 in a second tier, with pTau231, Aβ42/Aβ40 and NfL performing worse in terms of saved scans. For example, screening for a visually positive tau-PET scan with plasma pTau217 reducing 50.2% scans at 90% sensitivity, 41.6% at 95% sensitivity and saving 30.0% of scans at 97.5% sensitivity.”*

Results regarding PPV’s (Page 7 Line 16): *“When repeating the PPV analysis using tau-PET positivity determined with the visual read method in BioFINDER-2, pTau217 was again the only biomarker to lead to significant increases over the population rate of visual tau-PET positivity (45.3%). At 95% sensitivity, pTau217, showed a very similar PPV (66.5%, 95% CI 53.9-77.0%; 21.2% increase over population rate) as that of the quantitative-cutoff analysis above. At 90% and 97.5% sensitivities, pTau217 showed PPV’s of 74.0% (95% CI 63.5-81.9%) and 57.3% (95% CI 46.2-70.9%), respectively.”*

With regards to the remark “Indeed, the authors show that pTau217 has a PPV of 71%, but this also means that 29% of those who were pTau217-positive were tau-PET-negative. What is driving this discrepancy?”. This “discrepancy” comes from the fact that the screening plasma cutoffs evaluated were derived prioritizing high sensitivities. Even a very good test for a given outcome will not yield very high PPVs on a 37% prevalence (rate of tau-positivity in the cohort) if only sensitivity is prioritized. Had the screening cutoffs been derived with higher specificities, very high PPVs would have been achieved, but a lot of tau-PET-positive individuals would have been missed and denied a scan.

Remark #2: Related to this concern, the authors show that 10% of their AD group are tau-PET-negative. Do these 10% represent misdiagnoses or do these individuals have atypical tau PET patterns that are not captured with their temporal meta-ROI?

Response: We thank the reviewer for this comment. Among these n=15 (10%) individuals with an AD diagnosis who were tau-PET negative, n=9 were classified as tau-PET positive with the visual read. More

specifically, these 9 individuals had an “early AD pattern” in their visual read indicating a lower uptake and restricted to the medial temporal lobe. Among the 6 that were negative, 4 were visually negative and only 2 had an inconclusive visual read. The latter group (n=2) could have had a more atypical pattern of tau deposition, but given the very low frequency (n=2/548; 0.04% of the BioFINDER-2 included population), we do not consider this to be clinically significant within the study scope. Of note, all 125 AD patients who were positive with the quantitative cutoff were also positive with the visual read. The tau-PET radiotracer used in TRIAD has also demonstrated to be effective in capturing tau positivity in atypical AD (Therriault et al., Neurology; PMID: 33443136).

Remark #3: Another conceptual concern I have is that given that some studies have suggested that pTau changes occur earlier than tau-PET changes, is it more beneficial to use pTau to screen out those who are not tau-PET positive, rather than using it as a screen-in tool? In a clinical context, the benefit of confirming tau involvement using plasma pTau and then tau-PET is unclear to me (though of course it’s interesting from a research and clinical trial perspective, but the title says “in memory clinic settings”), whereas I can imagine it would be helpful to rule out tau involvement for a clinical patient.

Response: We thank the reviewer for the opportunity to clarify this. In our study, the goal was to improve the *prognostic* work-up of AD by proposing a strategy to optimize patient selection for referral to a tau PET scan, to make sure that tau-PET scans are only done in individuals where it will provide relevant prognostic information and thereby reducing costs and exposure to radiation. In the evaluated scenarios, such as the 95% sensitivity strategy with pTau217, 95% of all tau-PET-positive patients would indeed be referred to a tau PET scan (resulting in only 5% false negative cases based on plasma pTau217), while achieving a PPV of 67%. Importantly, in this scenario, the NPVs (the ability to rule-out tau involvement, as referred by the reviewer) were also very high (95.5% for pTau217).

To make this clearer we have now also added the PPVs and NPVs in a supplementary table to make it clear that most patients with tau pathology can be offered a scan when screening with pTau217, while still very accurately ruling out tau pathology in those individuals not referred to the scan.

The following tables represent NPVs and PPVs for all biomarkers and tested strategies in BioFINDER-2 and TRIAD.

Supp. Table 1. Negative and positive predictive values for each biomarker and screening strategy in BioFINDER-2.

Biomarker	Sensitivity	Negative Predictive Value (NPV)	Positive Predictive Value (PPV)
A β 42/A β 40	0.9	85.1% (77.9-88.8%)	45.0% (38.5-51.5%)
	0.95	89.3% (81.2-93.6%)	42.5% (36.4-49.7%)
	0.975	93.1% (83.3-97.1%)	40.6% (33.7-47.7%)
GFAP	0.9	89.9% (85.0-93.0%)	50.0% (40.2-60.1%)
	0.95	93.0% (86.6-95.9%)	44.9% (37.4-53.7%)
	0.975	94.4% (85.7-97.5%)	40.6% (33.7-49.4%)
NfL	0.9	81.6% (66.7-88.1%)	39.8% (33.2-47.9%)
	0.95	86.4% (72.0-92.3%)	38.8% (33.1-45.8%)
	0.975	90.9% (66.7-95.6%)	37.9% (32.5-43.9%)
pTau181	0.9	88.5% (82.9-92.7%)	48.6% (39.4-58.2%)
	0.95	91.7% (84.2-95.8%)	44.5% (36.5-55.4%)
	0.975	94.7% (88.2-97.4%)	42.0% (34.4-51.1%)
pTau217	0.9	92.5% (89.9-94.0%)	73.1% (62.2-83.1%)
	0.95	95.5% (93.3-97.3%)	67.3% (52.6-77.0%)
	0.975	97.9% (96.4-98.5%)	60.1% (45.2-72.7%)
pTau231	0.9	86.0% (79.7-90.6%)	45.1% (38.2-52.6%)
	0.95	89.5% (75.0-94.3%)	41.9% (36.1-48.0%)
	0.975	92.2% (75.0-96.8%)	39.7% (33.8-47.0%)

Supplementary Table 2. Negative and positive predictive values for each biomarker and screening strategy in TRIAD.

Biomarker	Sensitivity	Negative Predictive Value (NPV)	Positive Predictive Value (PPV)
A β 42/A β 40	0.9	66.7% (0.0-86.7%)	41.1% (26.5-53.9%)
	0.95	66.0% (0.0-85.7%)	40.0% (26.5-53.0%)
	0.975	68.0% (0.0-86.7%)	40.0% (26.5-53.1%)
GFAP	0.9	87.5% (77.3-92.6%)	58.6% (47.4-71.1%)
	0.95	92.3% (85.7-95.4%)	54.3% (43.0-69.5%)
	0.975	90.9% (83.3-94.7%)	52.9% (42.9-64.5%)
NfL	0.9	71.4% (33.3-85.2%)	50.0% (34.2-60.0%)
	0.95	66.7% (0.0-83.3%)	47.1% (34.1-59.3%)
	0.975	50.0% (0.0-83.3%)	47.0% (34.1-58.7%)
pTau181	0.9	87.5% (75.1-92.9%)	60.0% (42.1-75.0%)
	0.95	90.0% (57.9-94.9%)	54.4% (36.5-69.4%)
	0.975	87.5% (0.0-94.9%)	52.5% (36.5-68.4%)
pTau217	0.9	93.3% (88.9-96.2%)	86.2% (63.2-100.0%)
	0.95	96.2% (94.4-97.1%)	77.1% (60.1-94.6%)
	0.975	96.0% (94.4-97.0%)	73.1% (56.4-92.7%)
pTau231	0.9	86.7% (71.3-92.3%)	59.0% (44.5-72.1%)
	0.95	88.9% (75.0-94.9%)	51.8% (37.3-65.6%)
	0.975	84.5% (66.7-94.6%)	50.0% (36.2-62.7%)

Remark #4: An important detail that is missing is the time lag between the blood draw and the tau-PET scans.

Response: We thank the reviewer for this comment. In BioFINDER-2, the blood draw and tau-PET scan occur within the same visit or with a lag on the order of a few days.

In TRIAD, there is a lag between blood draw and scan, and this has been reported in the methods (Page 18 Line 22): “*There was no considerable time lag between tau-PET scanning and blood collection in BioFINDER-2, and in TRIAD this lag was low (0.43 years on average).*”.

Remark #5: Cognition here was only measured with the MMSE, which is not a comprehensive assessment of cognitive ability. This is an important limitation to consider.

Response: We thank the reviewer for this comment. Indeed, MMSE might not exhibit the highest sensitivity as a cognitive test; however, it has consistently demonstrated effectiveness at the group level in observational studies that include patients exhibiting cognitive symptoms. This is evident from the results of studies such as BioFINDER (Mattsson-Carlgrén et al, JAMA Neuro 2023, PMID: 36745413; Lezy et al, JAMA Neuro 2021, PMID: 34928318) and ADNI (Donohue et al, JAMA 2017, PMID: 28609533). The primary objective of the prognostic analyses conducted was to validate the efficacy of the proposed referral workflow in accurately identifying and excluding individuals in a prognostically meaningful manner, which is shown in our prognostic analyses (where cognitive decline was more effectively predicted by tau-PET only individuals theoretically referred to a tau-PET scan). As a result, we decided to maintain the change in MMSE as the principal outcome measure.

We include below a plot displaying the raw cognitive trajectories for each of the subpopulations included in our prognostic analyses. This shows that MMSE captures a clear signal of cognitive decline in this particular patient subset:

Remark #6: A minor point – I think at least one of the references is incorrect. Reference 24 is listed as the reference for the temporal meta-ROI definition but Cullen et al. does not include PET data.

Response: We thank the reviewer for noticing this. Indeed, it was a problem caused by changing reference manager software during drafting, and it has been fixed. The correct reference was Leuzy et al JAMA Neurol PMID: 32391858.

REVIEWERS' COMMENTS

Reviewer #1 (Remarks to the Author):

I am really impressed with the level of response and consideration from the authors, and I apologize for some comments (age of cohort and RO PET tracer) that were not accurate. I sincerely appreciate the careful responses as I was really able to learn at the same time.

One remaining issue, however, is I do agree with Reviewer 2 and 3 that MMSE is not the best cognitive test to have as the primary outcome. I would encourage a composite or some other cognitive test (i.e., verbal memory) right alongside the MMSE in the manuscript, or in replacement of the MMSE. If the MMSE is included, showing the raw MMSE trajectories would be ideal for the supplemental materials.

Reviewer #2 (Remarks to the Author):

The authors addressed all my comments and I believe that the manuscript can be accepted in its current form.

Reviewer #3 (Remarks to the Author):

The authors have sufficiently addressed all of my initial concerns. Additionally, their addition of TRIAD data as well as their sensitivity analyses using visual reads in BioFinder have significantly strengthened their well-written manuscript. I have just a few small questions/suggestions given their recent additions:

- One main strength of the visual reads sensitivity analyses is that the visual reads protocol allows for characterization of tau outside of the temporal-ROI. This is not highlighted in the manuscript and a reader would have to carefully read Smith et al. 2023 to know this. I think it's worth highlighting this point in the present manuscript.

- Figure 3 is very helpful and given the authors' recent addition of visual read information, it would be helpful to show a similar plot with visual reads - something like a bar plot showing the percentage of (1) normal image, (2) confined to temporal lobes, (3) widespread retention reaching parietal, occipital, or frontal lobes, and (4) inconclusive scans for negative and positive plasma pTau217 screening statuses.

- It is currently unclear if the "Tau-PET prognostic value in screened-in and screened-out patients" section and Figure 4 analyses were done in BioFinder only, or if these analyses collapsed BioFinder and TRIAD data. If the analyses predicting cognitive decline were done in BioFinder only because TRIAD data

did not have longitudinal MMSE available, this is worth mentioning in the limitations (or if there was some other reason, it's also worth clarifying why).

Reply to the reviewers' comments:

We are grateful for the time taken by the reviewers in carefully reading our manuscript and for their constructive suggestions and comments, as well as to the editorial members. Below, we address all of the reviewers' suggestions with results supporting the study's findings.

Please find below a detailed response to each reviewer in blue, with significant changes and additions made in the revised article document also marked.

Reviewer #1:

I am really impressed with the level of response and consideration from the authors, and I apologize for some comments (age of cohort and RO PET tracer) that were not accurate. I sincerely appreciate the careful responses as I was really able to learn at the same time.

Remark #1: One remaining issue, however, is I do agree with Reviewer 2 and 3 that MMSE is not the best cognitive test to have as the primary outcome. I would encourage a composite or some other cognitive test (i.e., verbal memory) right alongside the MMSE in the manuscript, or in replacement of the MMSE. If the MMSE is included, showing the raw MMSE trajectories would be ideal for the supplemental materials.

Response: We thank the reviewer for the positive feedback on the revisions and for the comments.

We have opted to maintain MMSE as the main cognitive outcome for the prognostic analyses. After careful consideration, we have decided to continue using the MMSE as the main cognitive outcome for our prognostic analyses. This decision is supported by data presented in Supplementary Figure 9 and detailed below, illustrating that MMSE is sufficiently sensitive to detect meaningful cognitive decline in both the SCD-MCI and all-cause dementia subgroups. Our findings are consistent with previous observational studies and clinical trials where MMSE has been effectively utilized to capture cognitive decline in symptomatic populations. Notably, in the recent successful phase 3 trials of lecanemab and donanemab, MMSE was included as a pre-specified secondary cognitive outcome showing positive results (van Dyck et al., NEJM, 2022; Sims et al., JAMA, 2023).

We have now included raw MMSE trajectories in Supplementary Figure 9, which were presented in the last revisions and suggested by the reviewer to be included in the Supplement in case we chose to maintain MMSE.

Supplementary Figure 9. Raw cognitive trajectories of prognostic analyses based on the plasma p-tau217 referral cutoff.

This is mentioned in the manuscript (Page 11 Line 1): “*Raw trajectories for these subpopulations are shown in the supplement (Supp. Fig. 9)*”.

Reviewer #2:

The authors addressed all my comments and I believe that the manuscript can be accepted in its current form.

Response: We thank the reviewer for taking the time to carefully read our manuscript and for the positive feedback.

Reviewer #3:

The authors have sufficiently addressed all of my initial concerns. Additionally, their addition of TRIAD data as well as their sensitivity analyses using visual reads in BioFinder have significantly strengthened their well-written manuscript. I have just a few small questions/suggestions given their recent additions.

Remark #1: One main strength of the visual reads sensitivity analyses is that the visual reads protocol allows for characterization of tau outside of the temporal-ROI. This is not highlighted in the manuscript and a reader would have to carefully read Smith et al. 2023 to know this. I think it's worth highlighting this point in the present manuscript.

Response: We thank the reviewer for this good suggestion.

We have now mentioned this in the:

- Methods (Page 18 Line 10): *“Briefly, with this visual read method, patients are classified in (A) normal image; no discernible [18F]RO948 retention, (B) retention of [18F]RO948 confined to the temporal lobes, (C) more widespread retention of [18F]RO948, reaching into the parietal, occipital, or frontal lobes, and (D) inconclusive scan, and patients with the B and C patterns are herein considered as abnormal tau deposition”.*
- Results (Page 7 Line 18; given the journal’s results-first, a brief description is made): *“When repeating the PPV analysis using tau-PET positivity determined with the visual read method in BioFINDER-2, pTau217 was again the only biomarker to lead to significant increases over the population rate of visual tau-PET positivity (45.3%) (Supp. Figure 3). The tau-PET scans were considered positive when abnormal accumulation occurred either confined to the temporal lobe (early deposition pattern) or when accumulation occurred in areas outside the temporal lobe (e.g. frontal, parietal, occipital cortices; late deposition pattern)”.*
- Discussion (Page 11 Line 19): *“Results were notably similar across the cohorts, and were very similar when using a visual read for tau-PET positivity determination in BioFINDER-2. This is reassuring given that the visual read method has been validated to detect tau accumulation outside the temporal lobe, which further supports that our proposed workflow may not lead to a denial of scans for patients with atypical patterns of tau deposition”.*

Remark #2: Figure 3 is very helpful and given the authors’ recent addition of visual read information, it would be helpful to show a similar plot with visual reads - something like a bar plot showing the percentage of (1) normal image, (2) confined to temporal lobes, (3) widespread retention reaching parietal, occipital, or frontal lobes, and (4) inconclusive scans for negative and positive plasma pTau217 screening statuses.

Response: We thank the reviewer for this comment. The suggested analysis has been performed and is shown below and in Supplementary Figure 8, and the results show that the evaluated screening strategy worked even better in those individuals with widespread retention.

Supplementary Figure 8. A sensitivity-focused plasma p-tau217 screening strategy shows good accuracy for capturing visual read tau-positivity.

This figure is now mentioned in the results (Page 9 Line 5): “In the supplement, we show that the 95% sensitivity plasma p-tau217 screening strategy also led to reliable classification of BioFINDER-2 tau-positive patients based on visual read, especially those with an advanced deposition pattern (Supp. Fig. 8).”

Remark #3: It is currently unclear if the “Tau-PET prognostic value in screened-in and screened-out patients” section and Figure 4 analyses were done in BioFinder only, or if these analyses collapsed BioFinder and TRIAD data. If the analyses predicting cognitive decline were done in BioFinder only

because TRIAD data did not have longitudinal MMSE available, this is worth mentioning in the limitations (or if there was some other reason, it's also worth clarifying why).

Response: We thank the reviewer for the opportunity to clarify this. Indeed, these analyses were carried only in BioFINDER-2 and not in TRIAD due to insufficient follow-up visits.

We have now mentioned it in the results (Page 9 Line 11): “Finally, we determined whether our suggested approach consisting of screening with plasma pTau217, followed by tau-PET only in those with elevated plasma pTau217, would be valuable when predicting cognitive decline *in the BioFINDER-2 cohort (prognostic analyses not performed in TRIAD; see Methods)*”

In the discussion (Page 15 Line 25): “*Lastly, prognostic analyses were carried in BioFINDER-2 only, due to insufficient follow-up for the included participants from TRIAD.*”.

In the methods (Page 17 Line 17): “*Availability of plasma biomarker levels and a tau-PET scan was required for inclusion in the cross-sectional (BioFINDER-2 and TRIAD) and longitudinal analyses (BioFINDER-2 only), available data was also required for at least one follow-up visit with available data for the Mini-Mental State Exam (MMSE).*”